# Therapeutic strategy for spinal muscular atrophy by combining gene supplementation and genome editing

Fumiyuki Hatanaka[1,2], Keiichiro Suzuki [3,4,5], Kensaku Shojima [1,6], Jingting Yu [7], Yuta Takahashi[1,2], Akihisa Sakamoto[1], Javier Prieto[1], Maxim Shokhirev[7], Estrella Nuñez Delicado [8], Concepcion Rodriguez Esteban[1,2] & Juan Carlos Izpisua Belmonte [1,2] ✉

Defect in the *SMN1* gene causes spinal muscular atrophy (SMA), which shows loss of motor neurons, muscle weakness and atrophy. While current treatment strategies, including small molecules or viral vectors, have shown promise in improving motor function and survival, achieving a definitive and long-term correction of SMA's endogenous mutations and phenotypes remains highly challenging. We have previously developed a CRISPR-Cas9 based homology-independent targeted integration (HITI) strategy, enabling unidirectional DNA knock-in in both dividing and non-dividing cells in vivo. In this study, we demonstrated its utility by correcting an SMA mutation in mice. When combined with *Smn1* cDNA supplementation, it exhibited long-term therapeutic benefits in SMA mice. Our observations may provide new avenues for the long-term and efficient treatment of inherited diseases.

Spinal muscular atrophy (SMA) stands as a severe inherited neuromuscular disorder resulting from the deletion or mutation of survival motor neuron gene 1 (*SMN1*). *SMN1* plays a pivotal role in various cellular processes and pathways, with a well-explored function in small nuclear ribonucleoprotein (snRNP) assembly[1]. SMA is characterized by the degeneration of lower motor neurons, leading to muscle weakness and atrophy. Its incidence is observed in 1 in 14,848 newborns, making it the most prevalent genetic cause of infant mortality[2]. In humans, two forms of SMN exist. *SMN1* serves as the primary gene for SMN production. Conversely, *SMN2*, a paralog of *SMN1*, differs only a few nucleotides and generates a truncated unstable protein with low levels of full-length SMN compared to *SMN1*. Despite only 10–20% functionality in the *SMN2* gene product, an elevated copy number inversely correlates with disease severity in SMA patients[3]. Current therapeutic approaches for SMA involve antisense oligonucleotides (ASOs) targeting RNA splicing of the *SMN2* gene, leading to an increase in full-length mRNA levels and subsequently enhancing the production of functional SMN[4]. While promising, ASOs face challenges, such as requirement of repetitive administrations. Another alternative approach for SMA involves gene supplementation therapy, delivering a fully functional copy of the *SMN1* gene using self-complementary adeno-associated viral (AAV) vectors, commonly packaged with serotype 9 (scAAV9). Important work by Mendell et al. demonstrated improved motor function and extended survival in SMA patients through a single intravenous injection of high dose scAAV9-*SMN1*[5]. However, this gene supplementation therapy cannot assure permanent restoration of endogenous *SMN1* expression due to the episomal nature of AAV vectors, preventing genomic integration into the host genomic DNA. Consequently, new strategies for in situ correction of endogenous mutated

[1]Gene Expression Laboratory, Salk Institute for Biological Studies, La Jolla, CA 92037, USA. [2]Altos Labs, Inc., 5510 Morehouse Dr., Ste. 300, San Diego, CA 92121, USA. [3]Institute for Advanced Co-Creation Studies, Osaka University, Osaka 560-8531, Japan. [4]Graduate School of Engineering Science, Osaka University, Osaka 560-8531, Japan. [5]Graduate School of Frontier Bioscience, Osaka University, Osaka 565-0871, Japan. [6]Department of General Internal Medicine, Hyogo Medical University School of Medicine, Hyogo 663-8131, Japan. [7]Integrative Genomics and Bioinformatics Core, Salk Institute for Biological Studies, La Jolla, CA 92037, USA. [8]Universidad Catolica, San Antonio de Murcia, Campus de los Jeronimos, 135, 30107 Guadalupe, Spain. ✉e-mail: jcbelmonte@altoslabs.com

sequences are imperative for achieving efficient and long-term improvement in SMA.

The CRISPR-Cas9 technology is a powerful genome-editing system that holds promise for addressing common inherited diseases[6,7]. Notably, recent advances in base editing have shown efficacy in modifying *SMN2* to enhance SMN expression[8]. However, this strategy may not be universally effective, particularly in severe SMA cases characterized by a low copy number of endogenous *SMN2*[3]. In this context, the success of in situ gene correction of the *SMN1* mutation holds the potential to provide a permanent cure for all SMA patients. Yet, its realization has been hampered, primarily due to the inherent challenges in accessing motor neurons within the spinal cord. The inefficiency of genome editing in non-dividing cells, especially motor neurons, poses a substantial hurdle, given the reliance on the homology-directed repair (HDR) pathway in conventional gene-repair approaches. Additionally, the limited in vivo delivery of genome-editing tools into the spinal cord hampers achieving sufficient genome correction to alleviate the disease phenotypes. Recently, we developed a non-homologous end joining (NHEJ)-based strategy for the targeted integration of transgenes, homology-independent targeted integration (HITI), effective in both dividing and non-dividing cells[9,10]. Since NHEJ is active throughout the cell cycle in a variety of cell types, including neurons[11,12], the HITI technology emerges as a promising solution to current challenges in efficiently and durably ameliorating SMA phenotypes. Here, we introduce a novel approach, termed Gene-DUET, which combines gene cDNA supplementation with genome editing using HITI. Our study demonstrates significant advancements in long-term survival and amelioration of SMA phenotypes. Besides improving current therapies for SMA, our findings may hold potential implications for the treatment of various inherited diseases, particularly neurodegenerative and neuromuscular disorders.

## Results

### AAV-mediated in vivo genome editing in spinal cord

AAVs have gained prominence as widely employed vectors for therapeutic gene delivery. Their utility has extended to gene therapy clinical trials for conditions such as SMA, Duchenne muscular dystrophy (DMD), and X-linked myotubular myopathy[5,13,14]. Over the past few decades, engineered AAV capsids have been developed to facilitate effective gene delivery into diverse tissues, with tailored tropism achieved through current strategies[15]. Given the swift and severe phenotypes in SMA model mice[16], high AAV transduction is imperative. A recent report showed that engineered AAV-PHP.eB capsid enables efficient transduction in the central nervous system[17]. Then, we first compared AAV-PHP.eB with AAV9, commonly used in clinical gene therapy for SMA patients. Systemically delivering AAVs was chosen based on the requirement for *SMN* in neurons and peripheral organs for therapeutic efficacy in SMA model mice[18]. We injected GFP-expressing AAV (AAV-GFP) into neonatal wild-type (WT) mouse by intravenous injection and analyzed GFP expression in the brain, lung, heart, stomach, liver, spleen, pancreas, kidney, muscle, and spinal cord 2 weeks later (Fig. 1a, b and Supplementary Fig. 1a, b). Real-time quantitative reverse transcription PCR (qRT-PCR) showed that *Gfp* expression was statistically higher in the spinal cord and lower in the liver when using AAV-PHP.eB compared to AAV9 (Fig. 1c, d). The analyses of tissue sections showed that AAV-PHP.eB surpassed AAV9 in efficiency, particularly in the spinal cord and brain (Supplementary Fig. 1c–f). Notably, AAV-PHP.eB exhibited robust transduction of motor neurons, as evidenced by abundant co-localization with NeuN, a motor neural marker, in the spinal cord (Fig. 1e). These results demonstrate the advantage of AAV-PHP.eB for transduction into spinal motor neurons. To assess the frequency of in vivo targeted gene knock-in, we utilized Ai14 mice carrying the CAG promoter at the *Rosa26* loci (Fig. 1f)[9]. Since the HITI technology can perform in vivo gene knock-in in non-dividing cells represented by neural cells, we

tested the in vivo efficacy of HITI in the spinal cord by using Ai14-Cas9 mice expressing Cas9 constitutively. HITI-mediated GFP knock-in at the *Rosa26* locus downstream of the CAG promoter was observed in the nucleus of the motor neurons and liver when the AAV-PHP.eB-Ai14-HITI including guide RNA (gRNA) expression cassette, GFPNLS-pA donor sandwiched by Cas9/gRNA target sequence and mCherry reporter were delivered into Ai14-Cas9 mice at postnatal day 0.5 (P0.5) (Fig. 1g and Supplementary Fig. 1g, h). These results suggest that HITI-mediated genome editing is successful in the spinal motor neurons.

### Targeted in vivo gene correction of SMA mice via HITI

Next, to validate the potential of HITI technology for gene correction in SMA, we employed SMA mice ($SMN2^{+/+}$; $SMN\Delta7^{+/+}$; $Smn1^{-/-}$) as a disease model, characterized by lacZ reporter gene insertion disrupting the endogenous *Smn1* gene in exon 2 and harbor two transgenic alleles of human *SMN2*[19–21]. To prevent the deleterious effect of CRISPR-Cas9-induced insertions/deletions (indels) in the endogenous exon, we targeted intronic sequences upstream of exon2/lacZ in chromosome 13 (Fig. 2a). Previously, unexpected recombination events were observed within the shared sequences between the genome and the donor[22]. To avoid such an unexpected occurrence, we excluded the homologous sequence from the donor construct by integrating a segment of rat intron 1 of *Smn1* including the splicing acceptor together with codon-optimized mouse *Smn1* cDNA (exons 2–8) and rat 3′UTR. Consequently, the codon-optimized mouse *Smn1* cDNA (exons 2–8) and rat 3′UTR effectively encapsulate the transcriptional unit of the endogenous mouse *Smn1* gene after precise targeted gene knock-in. The constructed pAAV-*SMN1*-HITI vector contains intron 1 targeting gRNA expression cassette and *SMN1* donor sandwiched by two gRNA target sequences. Systemic delivery of AAV-PHP.eB-*SMN1*-HITI and AAV-PHP.eB-Cas9 to SMA mice at P0.5 (Fig. 2b) resulted in HITI-mediated gene knock-in detected by PCR amplification exclusively in treated tissues after 2 weeks (Fig. 2c). We also verified the corrected genome sequences in amplicons by Sanger sequencing (Supplementary Fig. 2a). Importantly, HITI-treated SMA mice exhibited phenotypic improvement compared to untreated SMA mice. HITI-treated SMA mice demonstrated independent walking capacity at 2 weeks, contrasting with the inability of untreated SMA mice to stand (Fig. 2d and Supplementary Movies S1, 2). HITI substantially and significantly increased body weight and mean survival compared to untreated SMA mice, albeit the effect size was relatively modest. (Fig. 2e, f and Supplementary Fig. 2b). However, while HITI successfully edited the genome of SMA mice and rescued SMA phenotypes, it was insufficient for sustained therapeutic effects, with mice not surviving beyond 3 weeks despite significant improvements in behavior and survival analyses. These results suggest that HITI-mediated genome editing is successful in SMA mice and rescues SMA phenotypes but is not sufficient for a therapeutic strategy for SMA.

### Gene-DUET strategy through gene supplementation and genome editing

SMA mice displayed significantly lower body weight at birth compared to WT and heterozygous litters, indicating advanced disease phenotypes at the time of treatment (Supplementary Fig. 2c). The timing of gene correction by HITI was deemed late for effective SMA rescue, highlighting the importance of early treatment and high *SMN1* expression[18,23]. To address this, a new strategy, Gene-DUET, was devised, combining wild-type cDNA supplementation and genome editing. Wild-type cDNA supplementation was accomplished by over-expressing mouse *Smn1* cDNA as previous reports[24]. We redesigned the AAV vector (AAV-*SMN1*-DUET) containing the mouse *Smn1* coding sequence (CDS) under the CMV promoter and *SMN1*-HITI as used in the previous construct in Fig. 2a. Similar to current gene supplementation therapy, only mouse *Smn1* CDS would be expressed in the absence of

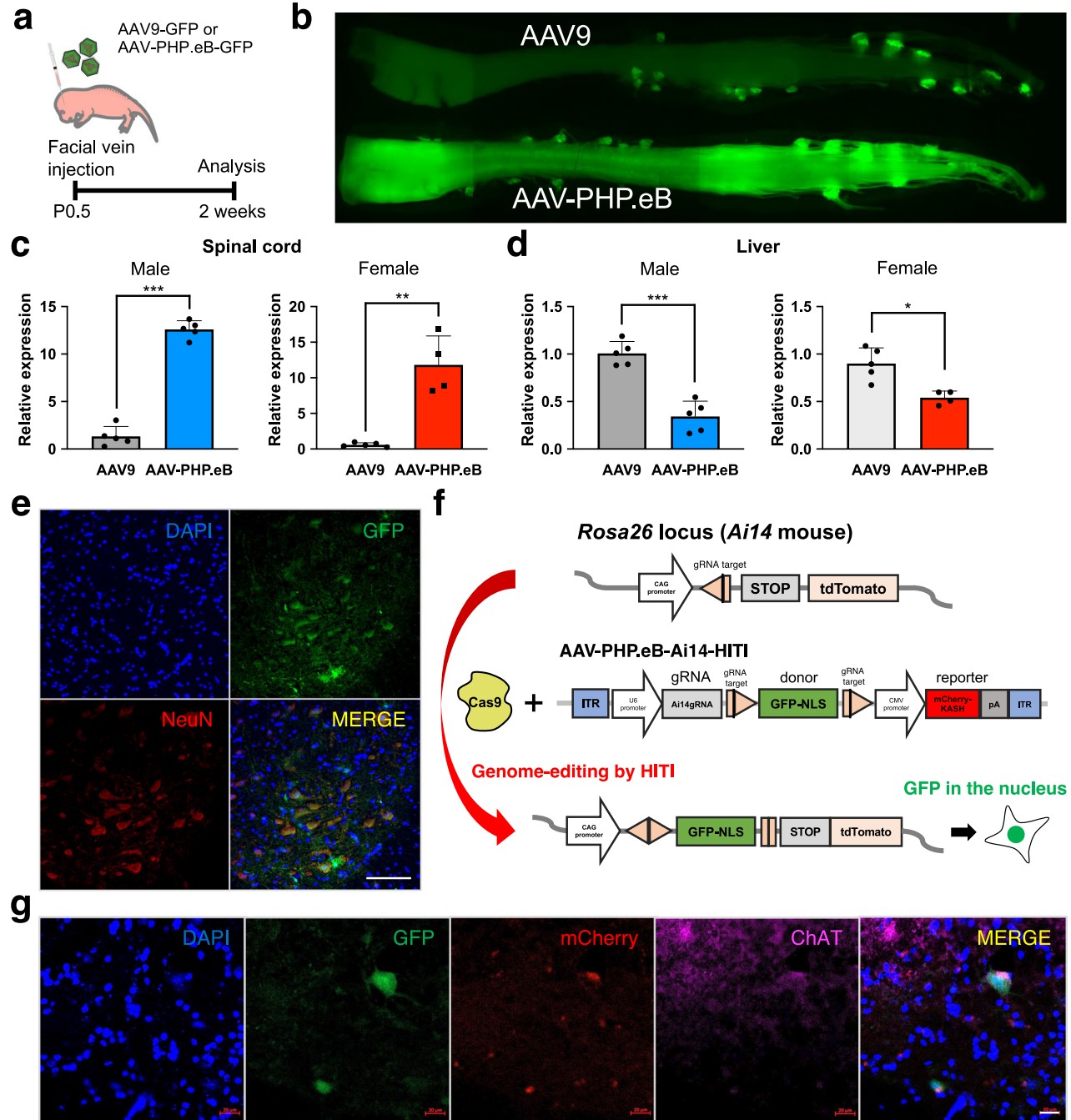

**Fig. 1 | AAV-mediated in vivo genome editing in spinal cord. a** AAV-GFP reporter ($1 \times 10^{11}$ GC) was administered via facial vein injection at P0.5 in WT mice. **b** GFP expression in the spinal cord of AAV9-GFP and AAV-PHP.eB-GFP injected mice. **c**, **d** RT-qPCR analysis for the expression of *Gfp* in the spinal cords (**c**) and livers (**d**) in AAV9-GFP and AAV-PHP.eB-GFP injected mice 3 weeks after the injections. Male mice (AAV9-GFP, $n = 5$; AAV-PHP.eB-GFP, $n = 5$) are shown on the left and female mice (AAV9-GFP, $n = 5$; AAV-PHP.eB-GFP, $n = 4$) are shown on the right. Data are represented as mean ± S.D. ***$p < 0.0001$, **$p = 0.0004$, *$p = 0.0051$, a two-sided unpaired Student's *t* test. **e** Representative fluorescent imaging of the spinal cord section in AAV-PHP.eB-GFP injected mouse. Scale bar, 1 mm. **f** Schematic of AAV construct for knock-in GFP-NLS downstream of the CAG promoter in the *Rosa26* locus of Ai14 mouse. Pink pentagon, Cas9/gRNA target sequence. Black line within the pink pentagon, Cas9 cleavage site. **g** Representative immunofluorescence images of GFP in the AAV-injected spinal cord section. The GFP signal in the nuclei was merged with mCherry and ChAT. Scale bar, 20 μm. Source data are provided as a Source Data file.

Cas9 (Fig. 3a). In contrast, mouse *Smn1* CDS and the gene-corrected *Smn1/SMN1* fusion gene can be co-expressed in the presence of Cas9 (Fig. 3b). Systemic delivery of AAV-PHP.eB-*SMN1*-DUET with or without AAV-PHP.eB-Cas9 was conducted in SMA mice at P0.5 (Fig. 3c and Supplementary Fig. 3a). Both cDNA-alone treatment (without Cas9) and DUET treatment (with Cas9) visibly improved the appearance of SMA mice compared to untreated SMA mice, as observed two weeks after the injections. (Figs. 2d, 3d and Supplementary Movies S1, 3, 4).

Tissue dissection revealed significantly enhanced sizes of the spinal cord, brain, heart, and muscle in both cDNA- and DUET-treated SMA mice (Fig. 3e, f and Supplementary Fig. 3b–f). Protein extracts from the spinal cords and brains of 2 weeks old mice were subjected to Western blot analysis to assess SMN1 expression (Fig. 3g). The blot revealed distinct bands corresponding to endogenous mouse SMN1 (37 kDa) in untreated WT mouse, which bands were not detected in untreated SMA mouse. The band of slightly higher molecular weight in SMA mice

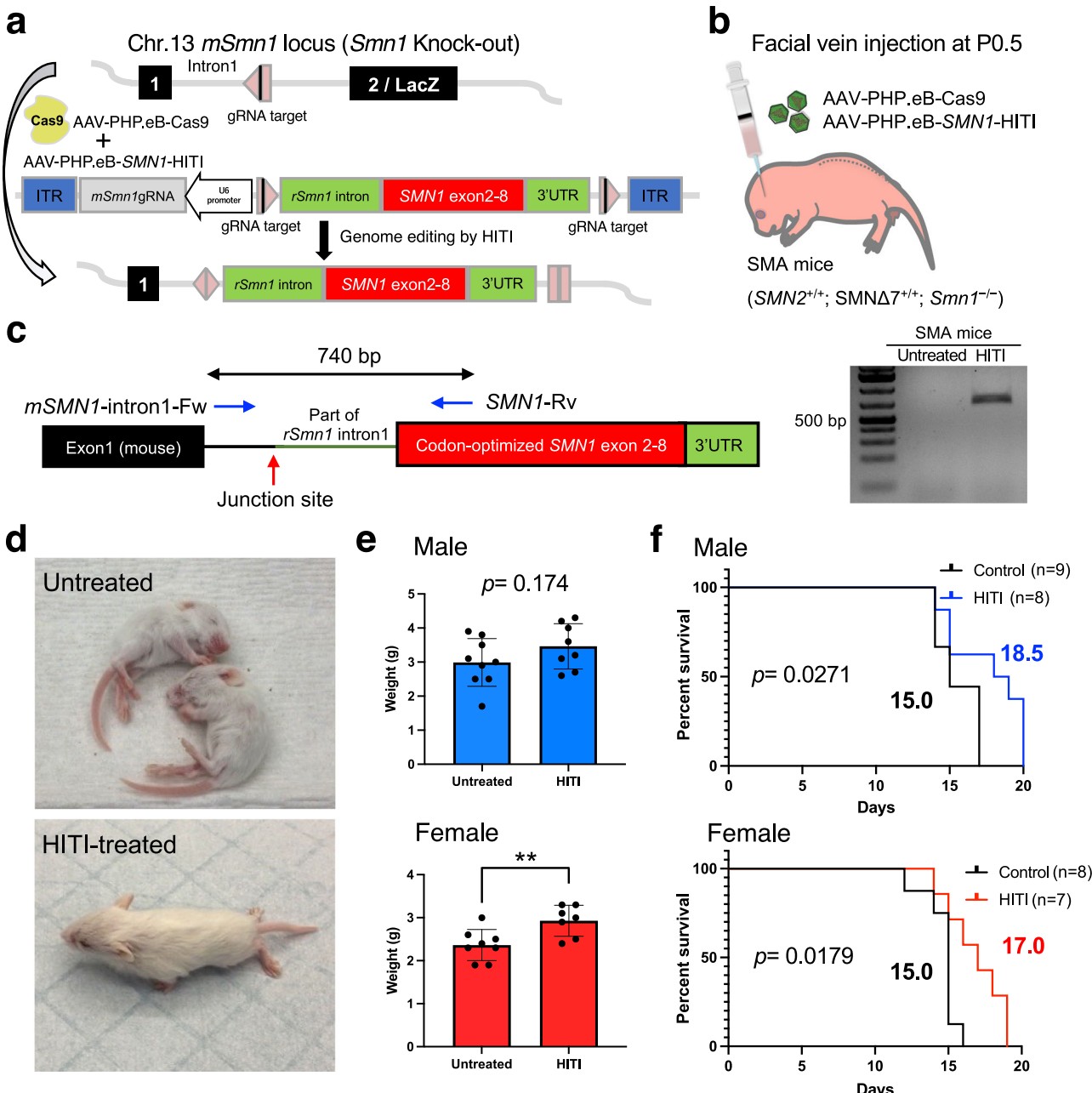

**Fig. 2 | HITI-mediated gene correction of SMA mice. a** Schematic of gene correction in SMA mice ($SMN2^{+/+}$; $SMN\Delta7^{+/+}$; $Smn1^{-/-}$). gRNA is expressed under the U6 promoter. Rat $Smn1$ intron, optimized mouse exon 2–8 and rat 3'UTR is sandwiched by Cas9/gRNA target sequence. The Cas9/gRNA creates a DSB at target sites and releases the donor. The donor sequence is integrated into the target site by NHEJ-based repair. Pink pentagon, Cas9/gRNA target sequence. Black line within the pink pentagon, Cas9 cleavage site. **b** AAVs ($1 \times 10^{11}$GC of each AAV) were systemically delivered via the facial vein in neonatal SMA mice. Body weight and life span were recorded after the injection. **c** The region between mouse intron1 and inserted rat intron1-optimized exon can be detected only in HITI-treated SMA sample by PCR. Blue arrows denote the location of the primers. Red arrow denotes the location of the junction site. **d** Gross morphology of untreated SMA mice (upper) and HITI-treated SMA mouse (lower) at 2 weeks old. **e** Body weight comparison between untreated and HITI-treated SMA mice of 12 days old with the males depicted on the top (Untreated, $n = 9$; HITI-treated, $n = 8$), and the females depicted on the bottom (Untreated, $n = 8$; HITI-treated, $n = 7$). Data are represented as mean ± S.D. \*\*$p = 0.0096$, a two-sided unpaired Student's $t$ test. **f** Survival rate comparison between untreated and HITI-treated SMA mice with the males depicted on the top, and the females depicted on the bottom. n is the number of animals per group. Log-rank (Mantel–Cox) test was used for survival curves. The $p$ value and median survival are indicated for all curves. Source data are provided as a Source Data file.

treated with cDNA and DUET indicates exogenous mouse SMN1 with a Myc-tag and FLAG-tag (40 kDa). Importantly, in SMA mice treated with DUET, the SMN band appeared at a position consistent with endogenous SMN1, suggesting HITI-mediated SMN1 expression (37 kDa). In addition, immunostaining confirmed the presence of SMN1 signals in the spinal cord of both cDNA- and DUET-treated SMA mice (Supplementary Fig. 3g). Improved motor function was evident in the righting reflex test, with significant enhancements in both male and female treated SMA mice 2 weeks after the injections (Fig. 3h). Some treated SMA mice exhibited ear and digital necrosis and shorter tails due to necrosis starting at 5 weeks of age until loss as reported in previous reports (Supplementary Fig. 3h)[25,26]. These results suggest that both cDNA and DUET treatments dramatically improve the phenotypes of SMA mice.

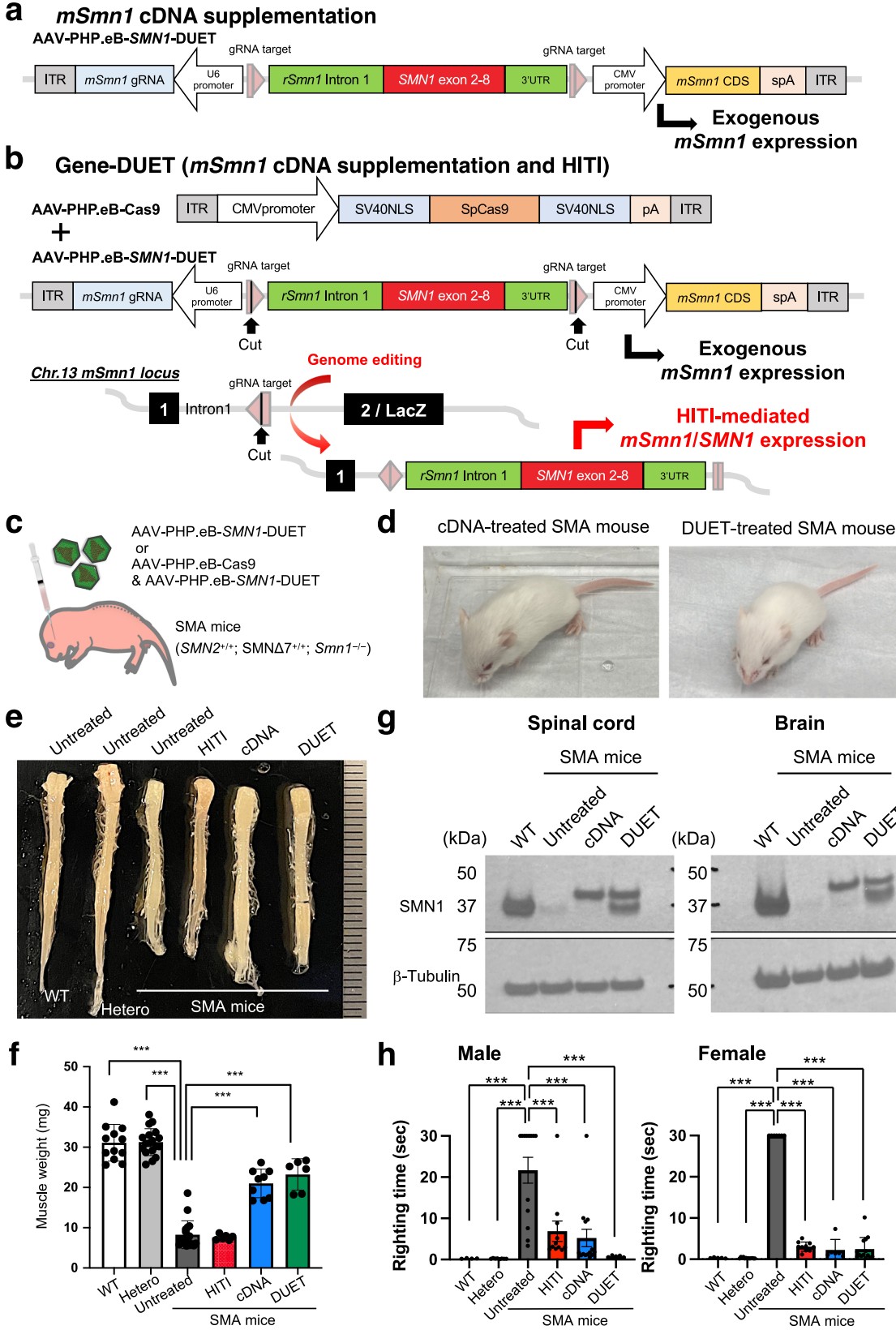

**Molecular correction via cDNA and DUET treatments in SMA mice**

We next performed RNA sequencing (RNA-seq) of spinal cord samples to understand the global transcriptional alterations by HITI, cDNA, and DUET treatments in SMA mice compared to untreated SMA mice and control heterozygous mice. Principal component analysis (PCA) and heatmap analyses clearly segregated untreated SMA mice from the healthy heterozygous mice (Fig. 4a and Supplementary Fig. 4a). The profile of cDNA and DUET treatments shifted SMA samples closer to heterozygous mice, suggesting a reversal of SMA-induced molecular dysfunction by the treatments. A functional gene enrichment analysis revealed that inflammatory pathways including p53 signaling pathway

**Fig. 3 | Gene-DUET strategy for SMA mice. a**, **b** Schematic representation of AAV-*SMN1*-DUET. Mouse *Smn1* (*mSmn1*) is expressed under CMV promoter. gRNA is expressed under the U6 promoter. Rat *Smn1* intron, optimized mouse exon 2–8 and rat 3'UTR is sandwiched by Cas9/gRNA target sequence. Pink pentagon, Cas9/gRNA target sequence. Black line within the pink pentagon, Cas9 cleavage site. Only *mSmn1* is expressed without Cas9 (**a**) and both *mSmn1* and gene-corrected *mSmn1*/*SMN1* are expressed in the presence of Cas9 (**b**). **c** AAVs ($1 \times 10^{11}$GC of AAV-*SMN1*-DUET or $2 \times 10^{11}$GC in total at an equal dose of AAV-*SMN1*-DUET and AAV-SpCas9) were systemically delivered via the facial vein in neonatal SMA mice. **d** Gross morphology of cDNA-treated SMA mouse (left) and DUET-treated SMA mouse (right) at 2 weeks old. **e** Gross morphology of the spinal cord in male WT, heterozygous, SMA mice with no treatment or each treatment at 2 weeks old. Scale bar, 1 mm. **f** Weight of the quadriceps femoris muscle in WT mice ($n = 12$), heterozygous mice ($n = 18$) and SMA mice with no treatment ($n = 18$) or HITI ($n = 6$), cDNA ($n = 9$) and DUET ($n = 6$) treatments. Both cDNA and DUET treatments improved

muscle atrophy in SMA mice. Data are represented as mean ± S.D. ***$p < 0.0001$, a two-sided unpaired Student's $t$ test (vs untreated SMA mice). **g** Western blot analyses were conducted to assess SMN1 expression. The upper band represents exogenous mSMN1 expression, and the lower band represents endogenous or HITI-mediated SMN1 expression in the spinal cords (left) and brains (right) of 2 weeks old WT and SMA mice with no treatment or subjected to cDNA and DUET treatments. WT mice were utilized as a reference for endogenous mouse SMN1 expression. β-Tubulin served as a loading control. **h** On the left, the righting ability of male WT mice ($n = 4$), heterozygous mice ($n = 8$), and SMA mice with no treatment ($n = 13$) or HITI ($n = 11$), cDNA ($n = 14$), and DUET ($n = 5$) treatments is depicted. On the right, the righting ability of female WT mice ($n = 5$), heterozygous mice ($n = 13$), and SMA mice with no treatment ($n = 7$) or HITI ($n = 10$), cDNA ($n = 6$), and DUET ($n = 14$) treatments is shown. Data are represented as mean ± S.E.M. ***$p < 0.005$, a two-sided unpaired Student's $t$ test. Source data are provided as a Source Data file.

**a**

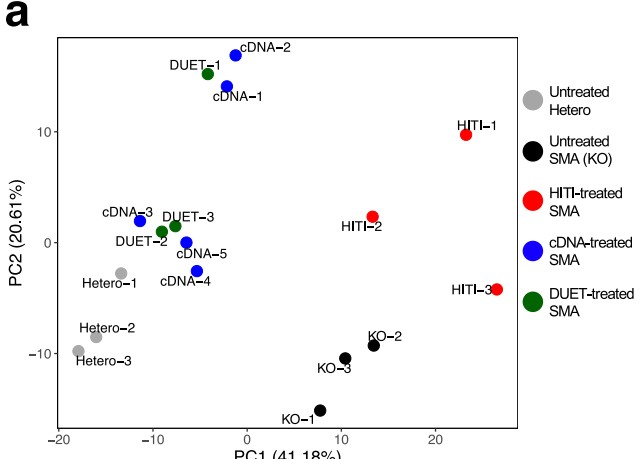

**b**

**Upregulation in SMA mice (vs Heterozygous mice)**

| Description | Genes | *p* value | FDR |
|---|---|---|---|
| Chemical carcinogenesis | 31 | 7.77E-16 | 2.49E-13 |
| Complement and coagulation cascades | 23 | 8.94E-10 | 4.10E-08 |
| Staphylococcus aureus infection | 15 | 5.59E-07 | 1.99E-05 |
| Cholesterol metabolism | 13 | 3.60E-06 | 9.63E-05 |
| Protein digestion and absorption | 18 | 4.38E-06 | 1.08E-04 |
| Bile secretion | 15 | 1.37E-05 | 3.15E-04 |
| p53 signaling pathway | 14 | 5.98E-05 | 1.13E-03 |
| Thyroid hormone synthesis | 14 | 8.23E-05 | 1.36E-03 |
| Metabolic pathways | 108 | 4.78E-04 | 6.67E-03 |
| Cytokine-cytokine receptor interaction | 29 | 5.30E-03 | 4.86E-02 |

**c**

**Downregulation in SMA mice (vs Heterozygous mice)**

| Description | Genes | *p* value | FDR |
|---|---|---|---|
| Alcoholism | 18 | 9.78E-06 | 3.14E-03 |
| Cholinergic synapse | 12 | 6.60E-05 | 1.06E-02 |
| Dilated cardiomyopathy (DCM) | 10 | 1.68E-04 | 1.80E-02 |
| Systemic lupus erythematosus | 12 | 6.10E-04 | 3.91E-02 |

**d**

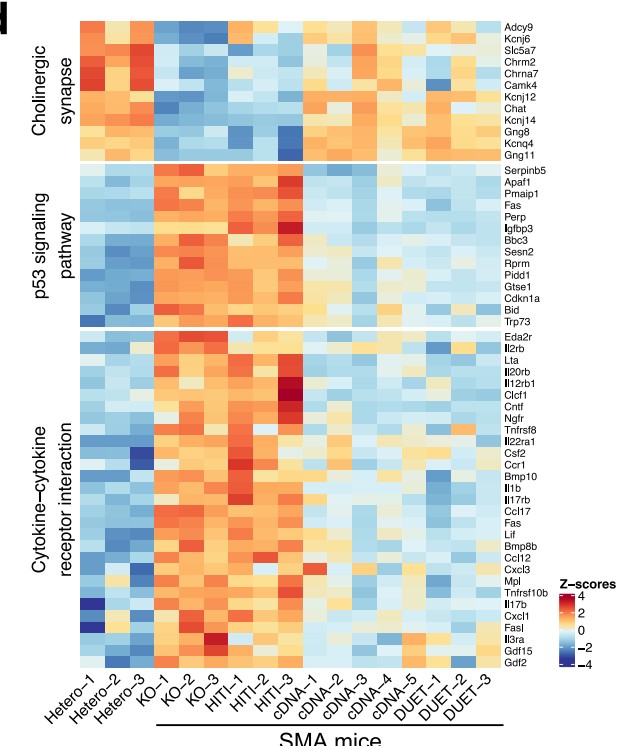

**Fig. 4 | Molecular correction via cDNA and DUET treatments in SMA mice. a** PCA analysis of RNA-seq in the spinal cords from untreated heterozygous mice (gray) and SMA mice with no treatment (black) or HITI (red), cDNA (blue) and DUET (green) treatments 2 weeks after the injections. **b**, **c** Pathways of upregulation (**b**) and downregulation (**c**) in SMA mice compared to heterozygous mice by gene-set enrichment analyses. Dysregulated genes were identified using the R package DESeq2, which employs the Wald test. The Wald test used in DESeq2 is inherently

two-sided, testing for both upregulated and downregulated genes. To account for multiple comparisons, the Benjamini–Hochberg method was applied to control the false discovery rate (FDR). **d** Heatmap for cholinergic synapse, p53 signaling pathway and cytokine-cytokine receptor interaction in the spinal cord from untreated heterozygous mice and SMA mice with no treatment or each treatment. Both cDNA and DUET treatments improved these molecular dysfunctions in the spinal cord of SMA mice.

and cytokine-cytokine receptor interaction were up-regulated while motor neuron pathway including cholinergic synapse was down-regulated in SMA mice compared to heterozygous mice (Fig. 4b, c and Supplementary Fig. 4b, c). These molecular changes were rectified by both cDNA and DUET treatments but not by HITI treatment. (Fig. 4d). Real-time qRT-PCR confirmed the significant repression of activated p53 downstream genes including *Gtse1*, *Ccng1*, *Perp* and *Sesn1* in cDNA- and DUET-treated spinal cord compared to untreated SMA samples (Supplementary Fig. 4d). These results suggest that both cDNA and DUET treatments reverse the molecular dysfunction in the spinal cord of SMA mice.

### Stable gene correction mediated by Gene-DUET

General appearance at 20 weeks old was similar between cDNA- and DUET-treated SMA mice (Fig. 5a). Both cDNA and DUET treatments led to a dramatic and significant increase in body weight compared to untreated SMA mice (Fig. 5b). Locomotor activity at 20 weeks old resembled that of heterozygous mice, indicating sustained motor function improvement (Supplementary Fig. 5). The concern with neonatal AAV treatment is the loss of the transgene along with cell division during tissue growth. The previous report showed quick reduction of vector genome copies in the liver over a few weeks after neonatal AAV treatment[27]. Indeed, GFP expression in the spinal cord declined after 1 year compared to 2 weeks in AAV-PHP.eB-GFP injected WT mice (Fig. 1b and Supplementary Fig. 6a). We also found a significant reduction in cDNA-derived exogenous mouse *Smn1* expression in the spinal cord and liver at 1 year old compared to at 2 weeks old in cDNA-treated heterozygous mice (Supplementary Fig. 6b, c). In theory, genome editing by HITI might offer more advantages than cDNA supplementation for persistent SMN expression. To examine the efficiency of gene correction in DUET-treated SMA mice, we extracted genomic DNA from the spinal cords, brains, and livers of the treated mice and enriched for target regions by using customized probes that could hybridize and pull down the genomes around the Cas9 cleavage site (Fig. 5c). Gene correction by DUET was evaluated at 20 and 40 weeks old through deep sequencing, revealing detectable and stable corrected sequences in all tested tissues (Fig. 5d, e). These results suggest that gene correction by Gene-DUET strategy is stable for a long time in SMA mice. Crucially, both cDNA and DUET treatments significantly enhanced the survival of SMA mice compared to untreated SMA mice. More importantly, DUET-treated SMA mice demonstrated improved survival over cDNA-treated SMA mice, especially in males (Fig. 5f, g). These data suggest the synergistic effect of the Gene-DUET strategy by gene supplementation and genome editing.

## Discussion

Gene therapy medicine, onasemnogene abeparvovec-xioi (Zolgensma), for treating SMA by cDNA supplementation, was approved by the FDA in 2019. Its effectiveness has been demonstrated for up to 7 years in a small cohort of children and for over 3 years in a larger group, with no reported decline in clinical effect. However, the long-term efficacy of this treatment is yet to be fully determined. The development of HITI technology has enabled the correction of genomic mutations in non-dividing cells through the NHEJ pathway, and here we provide a first demonstration of its utility in an SMA model in mice. Unlike cDNA supplementation, this in situ gene correction approach ensures stable and permanent *Smn1* expression. While gene correction alone couldn't rectify all phenotypic alterations due to immediate and extensive changes after birth, it did address a subset of phenotypic changes in SMA mice. Combining HITI-mediated genome editing and gene supplementation exhibited increased cell and tissue phenotypic restoration, along with the increase in SMN1 levels resulting from our strategy, with contributions from both cDNA supplementation and HITI-mediated genome editing. Importantly, our Gene-DUET demonstrated a significant boost in survival time, especially in male mice. Previous reports also showed sex-specific amelioration of phenotype by antisense oligonucleotide treatment using mild SMA model mice[28,29]. In our study, one plausible factor is the size disparity between female and male SMA mice at the time of AAV injection, along with developmental differences, leading to differences in AAV density. We hypothesize that the higher AAV density in female SMA mice, despite receiving the same AAV dosage, could influence treatment efficiency. In fact, we also observed that the mean survival of cDNA-treated female SMA mice was two times longer than that of cDNA-treated male SMA mice.

AAV stability in neurons, as previously reported, is more pronounced than in proliferative cells[30]. However, our data indicated stable genome-editing rates in the spinal cord and an increase in the liver. These findings align with the concept that stable *SMN* expression, sustained by HITI-mediated *SMN1*, is crucial for both motor neurons and peripheral organs, providing a foundation for the observed additional therapeutic benefits with the Gene-DUET strategy. Recently, the combination of base editing with antisense oligonucleotide treatment showed promise in ameliorating SMA phenotypes[8], however, that may not be universally effective, especially in severe SMA cases. Additionally, another recent report showed that there was no apparent improvement in the phenotypes of SMA mice using a base-editing approach[31]. This is due to the rapid and severe disease progression of SMA mice. The choice of AAV-PHP.eB and systemic delivery was strategic due to its high transduction efficiency in the spinal cord and other organs of SMA mice, highlighting the success of the Gene-DUET strategy over conventional gene supplementation. However, some cDNA- or DUET- treated SMA mice exhibited necrosis due to limitations in systemic protection and peripheral tissue outcomes. Further AAV evolution with tropism, promoter engineering, or size limitation could be considered for more efficient transduction and practical translational applications.

Although gene therapy has shown significant effectiveness in treating SMA, there have been reports of risks including liver toxicity and neurotoxicity[30]. It's important to consider potential adverse effects, and further studies of Gene-DUET in other animal models, including non-human primates, should be required for clinical application.

In summary, our Gene-DUET strategy provides new exploratory avenues for the treatment of SMA in humans. This approach has great potential for the field of genome-editing technologies that may hold potential implications for the treatment of various inherited diseases, particularly neurodegenerative and neuromuscular disorders.

## Methods

### Plasmids

pAAV-CAG-GFP (addgene 37825) and pUCmini-iCAP-PHP.eB (addgene 103005) were purchased from Addgene. pAAV-CMVc-Cas9 (addgene 106431) was established previously[32]. To construct gRNA expression vectors, each 20 bp target sequence was sub-cloned into pCAGmCherry-gRNA (Addgene 87110). The CRISPR/Cas9 target sequences (20 bp target and 3 bp PAM sequence (underlined)) used in this study are shown as following: *Smn1* intron 1 targeting gRNA (GCCATACCATAAGACGACCG<u>AGG</u>). The downstream of CAG promoter in Ai14 mouse (<u>TAGGAACTTCTTAGGGCCCG</u><u>CGG</u>) gRNA expression plasmid has been established in the previous paper[9]. pAAV-nEFCas9 (Addgene 87115) and AAV-Ai14-HITI (Addgene 87117) were established in the previous paper[9]. To construct donor/gRNA AAV for HITI-mediated *Smn1* gene correction, part of intron 1 including splicing acceptor site, 3'UTR and downstream were amplified from rat genome isolated from Brown Norway rat. The mouse *Smn1* exon 2–8 was codon optimized and synthesized in IDT. The assembled fragment was sandwiched by two *Smn1* intron 1 gRNA target sequence and sub-cloned into between ITRs of PX552 purchased from Addgene (Addgene 60958), and generated pAAV-*SMN1*-HITI. To construct

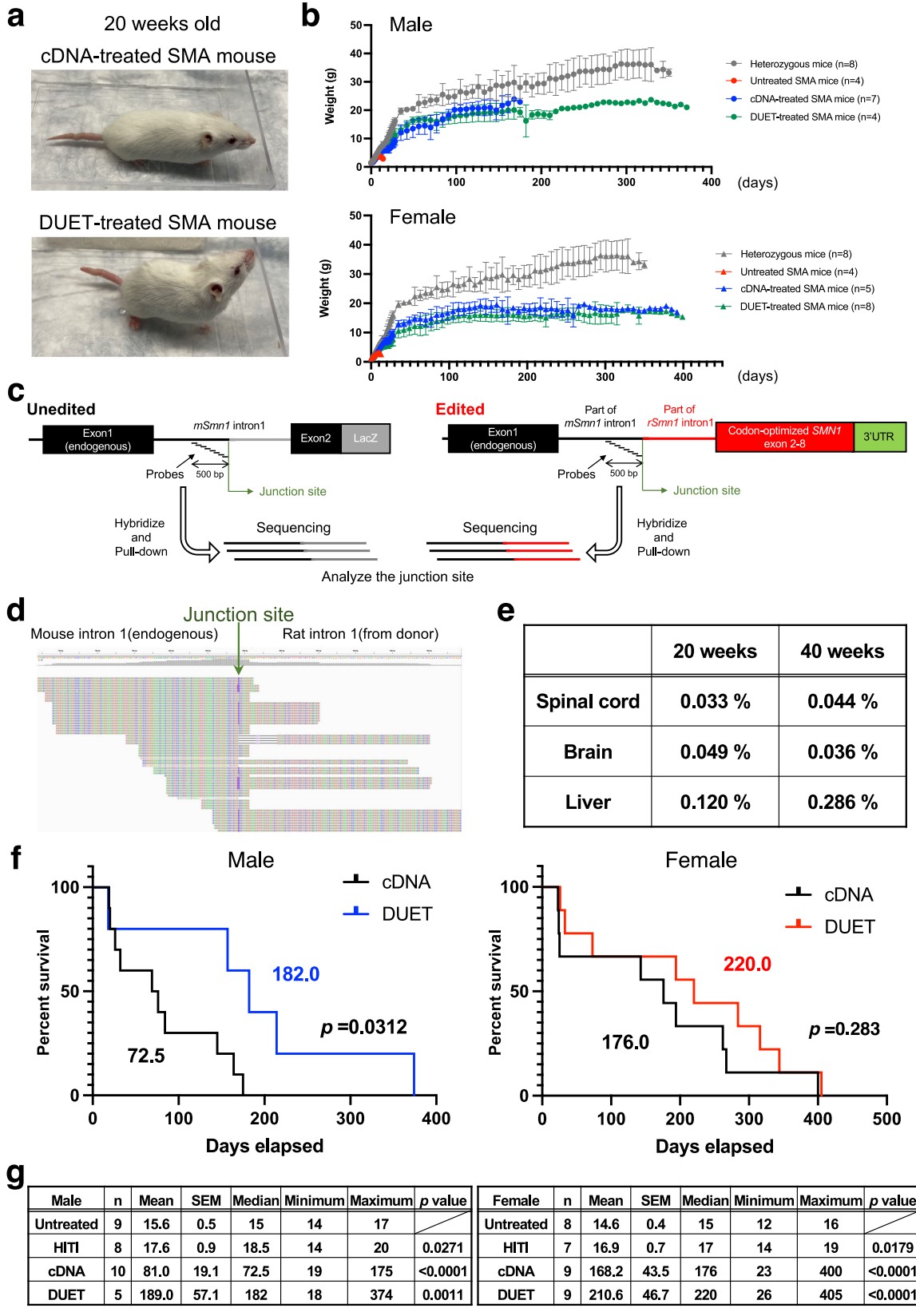

pAAV-*SMN1*-DUET, CMV-*mSmn1* CDS expression cassette (ORIGENE MR203917) was sub-cloned into pAAV-*SMN1*-HITI.

**Animals**

C57BL/6, Rosa-LSL-tdTomato (known as Ai14), Rosa26-Cas9 knock-in mice (Stock No: 024858), SMNΔ7 mice (FVB.Cg-*Grm7*<sup>Tg(SMN2)89Ahmb</sup> *Smn1*<sup>tm1Msd</sup> Tg(SMN2*delta7)4299Ahmb/J, Stock No: 005025) were purchased from the Jackson laboratory. We mated Ai14 mouse and Rosa26-Cas9 knock-in mouse to generate the Ai14-Cas9 mouse. The mice were housed in a 12-h light/dark cycle (light between 06:00 and 18:00) in a temperature-controlled room (22 ± 1 °C) with free access to water and food. All procedures were performed in accordance with

**Fig. 5 | Synergistic benefits of the Gene-DUET strategy. a** Gross morphology of cDNA-treated SMA mouse (upper) and DUET-treated SMA mouse (lower) at 20 weeks old. **b** Body weight through time in untreated heterozygous mice and SMA mice with no treatment or each treatment. Gray, red, blue and green line denotes untreated heterozygous mice, untreated SMA mice, cDNA-treated SMA mice and DUET-treated SMA mice respectively. Male mice (circles) are shown on the top and female mice (triangles) are shown on the bottom. n is the number of animals per group. **c** Schematic of the target enrichment-genome sequencing. Customized probes were designed by 500 bp upstream from the junction site. Green arrow indicates Cas9 cleavage site. **d** Screenshot of representative edited reads around the junction site (green arrow) in Integrative Genomics Viewer (IGV). **e** Target enrichment-genome sequencing analysis of the on-target genomic edits

compared to unedited genome in the spinal cords, brains and livers from 20 and 40 weeks old DUET-treated SMA mice. **f** Survival rate between cDNA- and DUET-treated SMA mice with males (left) and females (right). Log-rank (Mantel−Cox) test was used for survival curves. The *p* value and median survival are indicated for all curves. **g** Summary of survival analysis. Data of untreated or treated SMA mice are shown in the table. Male mice are shown on the left and female mice are shown on the right. n is the number of animals per group. The Mean, SEM, minimum and maximum of survival days per group are indicated. Log-rank (Mantel−Cox) test was used for statistical analysis between no treatment and each treatment. The test was conducted as a two-sided analysis, and no adjustments were made for multiple comparisons. Source data are provided as a Source Data file.

protocols approved by the IACUC and Animal Resources Department of the Salk Institute for Biological Studies.

## AAV production
AAV9 and AAV-PHP.eB viral particles were generated by or following the procedures of the Gene Transfer Targeting and Therapeutics Core at the Salk Institute for Biological Studies.

## Intravenous AAV injection
The newborn (P0.5) mice were used for intravenous AAV injection as following previous report[32]. The AAV mixtures ($1 \times 10^{11}$ GC of each AAV) were injected via facial vein of the newborn mice. After the injection, pups were allowed to completely recover on a warming pad and then returned to the home cage.

## Genome extraction
Genomic DNA was extracted from cells and tissue samples using DNeasy Blood & Tissue Kits (69506, Qiagen) following the manufacturer's instruction.

## RNA analysis
Total RNA was extracted from cells and tissue samples using either TRIzol (Invitrogen) or RNeasy Kit (Qiagen) followed by cDNA synthesis using Maxima H Minus cDNA Synthesis Master Mix (Thermo Fisher). Real-time qPCR was performed using SsoAdvanced SYBR Green Supermix and analyzed using a CFX384 Real-Time system (Bio-Rad). For the expression of *Gfp*, *Gtse1*, *Ccng1*, *Perp*, *Sesn1* and *Gapdh* with the following primers: *Gfp*-Fw ACGACGGCAACTACAAGACC, *Gfp*-Rv ACCT TGATGCCGTTCTTCTG, *Gtse1*-Fw TGACAAAGAGAACGTGGACTCAC, *Gtse1*-Rv GAGGTGGGAGGCTTAGGTTC, *Ccng1*-Fw TTATGGGACGTAA GGAGACACC, *Ccng1*-Rv ATGGTTCCAGCTACTCTAGGTTG, *Perp*-Fw TTTGGGAATGCGTGTCTCTG, *Perp*-Rv TCAACTGTCTTTGCAGCACC, *Sesn1*-Fw TTCTCTGAGCCTGGAGGACAG, *Sesn1*-Rv CTTCAAAGTCAGG GTCCCGA, *Gapdh*-Fw ACGGGAAGCTCACTGGCATGGCCTT, *Gapdh*-Rv CATGAGGTCCACCACCCTGTTGCTG. All analyses were normalized based on amplification of mouse *Gapdh*.

## Immunohistochemistry
Tissues were harvested after transcardial perfusion using ice-cold PBS, followed by ice-cold 4% paraformaldehyde in phosphate buffer for 15 min. Tissues were dissected out and postfixed in 4% paraformaldehyde overnight at 4 °C and cryoprotected in 30% sucrose overnight at 4 °C and embedded in OCT (Sakura Tissue-Tek) and frozen on dry ice. Serial sections at 12 μm were made with a cryostat and collected on Superfrost Plus slides (Fisher Scientific) and stored at −80 °C until use. Immunohistochemistry was performed as follows: sections were washed with PBS for 5 min 3 times, incubated with a blocking solution (PBS containing 2% donkey serum (or 5% BSA) and 0.3% Triton X-100) for 1 h, incubated with primary antibodies diluted in the blocking solution overnight at 4 °C, washed with PBST (0.2% Tween 20 in PBS) for 10 min 3 times, incubated with secondary antibodies conjugated to Alexa Fluor 488, Alexa Fluor 546, or Alexa Fluor 647 (Thermo Fisher)

for 1 h at room temperature. After washing, the sections were mounted with mounting medium (DAPI Fluoromount-G, SouthernBiotech). The primary antibodies used in this study were anti-GFP, 1:500 (Aveslabs); anti-NeuN, 1:100 (MCA-1B7, EnCor Biotechnology); anti-ChAT, 1:200 (NB110-89724, Novus Biologicals); anti-SMN, 1:200 (610647, BD Biosciences).

## Image capture and processing
Immunocytochemistry samples of mice samples were visualized by confocal microscopy using a Zeiss LSM 710 Laser Scanning Confocal Microscope (Zeiss). Images were processed by ZEN2 Black edition software (Zeiss).

## Western blot
The spinal cord and brain samples were homogenized using a Dounce homogenizer in lysis buffer (1× PBS, 1% NP40, 0.5% sodium deoxycholate, 0.1% SDS, with a protease inhibitor cocktail from Santa Cruz). The samples were then resolved on SDS-PAGE. Proteins were transferred to a polyvinylidene fluoride membrane. The membranes were blocked in TBST (10 mM Tris-HCl, pH 7.5, 150 mM NaCl, and 0.1% Tween 20) containing 5% milk and probed using the indicated primary antibodies conjugated with horseradish peroxidase. Specific protein bands were detected using the ECL Plus Western Blotting Substrate from Thermo Fisher. The following antibodies were used: anti-SMN (610647, BD Biosciences) and anti-β-Tubulin (MA5-11732, Invitrogen).

## Righting reflex test
The righting reflex of untreated or treated mice was compared at postnatal 14 days. Mice were laid on their back and the time needed to flip over was recorded, with a maximum of 30 s allowed. Three trials were performed for each mouse and the shortest time was used for analysis.

## Open field test
Mice were individually placed into clear Plexiglass boxes (40.6 × 40.6 ×38.1 cm) surrounded by multiple bands of photo beams and optical sensors that measure horizontal (ambulatory) and vertical (rearing) activity (Med Associates, USA). Mice movement was detected as breaks within the beam matrices and automatically recorded for 30 min.

## RNA sequencing and data analysis
RNA was extracted from the spinal cord and prepared for RNA sequencing with TruSeq Standard mRNA sample Prep Kit (Illumina). Deep sequencing was performed on the Illumina NovaSeq6000, at pair end 150 bp. Sequenced reads were quality-tested using FASTQC v0.11.8 (https://www.bioinfomatics.babraham.ac.uk/projects/fastqc/) and adapters were trimmed using Cutadapt v2.4[33] with parameters '-j 8 -f fastq -e 0.1 -q 20 -O 1 -a AGATCGGAAGAGC'. Paired reads after trimming were aligned to custom mouse transcriptome and quantified at transcript level using Kallisto v0.46.0[34] with arguments 'quant -b 30'.

The custom transcriptome index was built with 'kallisto index' by merging the cDNA FASTA files of mm10 mouse transcriptomes and three genome-editing conditions of gene *Smn1* (including cDNA, HITI and Gene-DUET). Raw gene expression was estimated and summarized from transcript-level abundance with R package, tximport v1.12.3. Differential gene expression was performed on the raw gene counts with the R package, DESeq2 v1.24.0, using replicates to compute within-group dispersion. Differentially expressed genes were defined as having a false discovery rate (FDR) < 0.05 and a |Fold Change| >1.5 when comparing two experimental conditions. Principle Component Analysis (PCA) was carried out on normalized gene counts using the R prcomp function. Heatmaps were generated with the R package, ComplexHeatmap v2.0.0. Overrepresentation Analysis (ORA) was performed with WebGestalt[35] using KEGG pathways with FDR < 0.05 as the significance threshold, protein coding genes as the reference list, a minimum number of genes in a category of 5, and visualizing as a barplot with normalized enrichment scores.

### Target enrichment-genome sequencing analysis

Target enrichment-genome sequencing was performed using the SureSelect XT HS2 DNA system for Illumina Multiplexed Sequencing (Agilent Technologies) with a customized capture library following manufacture's protocol. The customized probes to capture specific genomic loci were designed to include chr13:100125170-100125670 which 500 bp is upstream of Cas9 cleavage site. 50 ng of genomic DNA was fragmented by using SureSelect Enzymatic Fragmentation Kit. The DNA libraries were prepared by using SureSelect XT HS2 DNA Library preparation Kit and amplified by unique dual indexing primer pair following manufacture's protocol. After hybridization by customized probes, captured libraries were purified using streptavidin beads and amplified by PCR. Following indexing and sample pooling, sequencing was conducted using Illumina MiniSeq, at pair-end 150 bp. Sequenced reads were trimmed using cutadapt v2.4[33] with parameters '-j 8 -f fastq -e 0.1 -q 20 -O 1 -a AGATCGGAAGAGC' and then mapped to a custom genome reference with BWA v0.7.12[36] using default parameters. The custom reference consisted with DNA sequences of mouse chromosome 13 and the edit region around Cas9 cleavage site, upstream of which included the exon 1 and intron 1 of mouse *Smn1* and downstream of which included part of rat *SMN1* intron and optimized mouse exon 2–8. Resulting sam files were converted to sorted indexed bam files using samtools v1.9. Reads mapping to specific genomic region was obtained using 'samtools view' and then converted into bed format using bedtools v2.3 with arguments 'bamToBed -cigar', from which number of fully matched reads, reads with mismatch or indels were counted. Resulting counts were manually processed to produce position-specific figures and analyses.

### Statistics and reproducibility

All of the data are presented as the mean ± S.D. or S.E.M. Each experiment was repeated independently at least three times. Statistical parameters including statistical analysis, statistical significance in the Figure legends. For statistical comparison, Two-tailed Student's *t* test and Log-rank (Mantel–Cox) test for survival curves were performed by Prism 9 Software (GraphPad).

### Reporting summary

Further information on research design is available in the Nature Portfolio Reporting Summary linked to this article.

## Data availability

Data supporting the findings of this study are available within the paper and its supplementary information files. RNA-sequence data was deposited in the Gene Expression Omnibus under the accession number GSE207181. Source data are provided with this paper.

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

## Acknowledgements

We are grateful to Grace Chou, Tzu-Wen Wang, Ling Ouyang and Nasun Hah for next-generation sequencing; Animal Resources Department for animal care; M. Schwarz and P. Schwarz for administrative help. F.H. was partially supported by the Uehara Memorial Foundation and the Leave a Nest Research Grant. K.Shojima was partially supported by the Cell Science Research Foundation, the Uehara Memorial Foundation and Hashimoto Municipal Hospital. This work was supported by UCAM, the Japanese Society for the Promotion of Science KAKENHI, 21H04811 and AMED, JP22ek0109521.

## Author contributions

F.H. K.Suzuki and J.C.I.B. designed all experiments. F.H. performed and analyzed all experiments. K.Suzuki designed the constructs. K.Shojima, A.S. and J.P. produced AAVs. K.Shojima, Y.T. C.R.E and E.N.-D supervised and analyzed in vivo experiments. J.Y. and M.S. provided suggestions and analyzed RNA-seq/Genome-seq. F.H. wrote the manuscript. K.Suzuki and J.C.I.B. provided feedback and reviewed the manuscript. All authors contributed to editing the manuscript.

## Competing interests

F.H., Y.T., C.R.E and J.C.I.B. are employees of Altos labs. The other authors declare no competing interests.
