## [Peer Review File · Nature Communications]

Therapeutic strategy for spinal muscular atrophy by combining gene supplementation and genome editingREVIEWER COMMENTS

Reviewer #1 (Remarks to the Author):

In this study the authors sought to determine if a CRISPR-Cas9 based homology-independent targeted integration strategy that targets the mouse smn gene in combination with delivery of human SMN1 would improve the SMA disease phenotype using a severe SMA mouse model. This strategy would be beneficial over the approved gene therapy strategy which suffers from a dilution effect over time in dividing cells. While the SMA gene therapy treatment has dramatically changed the disease landscape, treated patients have shown signs of motor decline within 6 months to a year after treatment, requiring additional treatment with the splicing modifiers. Therefore, alternative strategies to provide more stable induction of SMN is warranted. The studies outlined here show promising results to this end, but there remain areas of concern that should be addressed before this study could be considered suitable for publication.

Major Concerns:

1. The language used throughout the manuscript is very imprecise. The manuscript requires extensive editing, particularly in the results section. Statements like "look very healthy" does not reflect quantitative assessment of improved disease phenotype.
2. The finding that AAV-PHP.eB was more efficient in the spinal cord and brain than AAV9 is not substantiated by the data presented here since there was only visual qualitative assessment and no quantification of that data.
3. Given that that DUET strategy was employed because of the potential lag in inducing SMN levels inherent in the HITI-mediated gene correction strategy, it would be pertinent to compare the difference in SMN protein levels across the tissues examined. Outside of some gene expression data in Supplemental Figure 6 there is nothing in this study that demonstrates persistent improvement in SMN levels, which was the central hypothesis.
4. The sex differences observed in the DUET treated animals are potentially interesting but this needs to be reconciled since there are not many sex differences that have been reported in SMA disease presentation or response to treatment. The only other sex

difference, this reviewer can remember, is in the milder SMN2b- mouse, not the SMN delta7 mice used in this study.

Minor Concerns:

1. Rats(rodents) have a single copy of the SMN gene. It's therefore unclear what the authors mean by "we removed the homologous sequence on the donor by incorporating a portion of rat intron 1 of Smn1 including the splicing acceptor, codon-optimized mouse Smn1 cDNA"
2. This statement is not factually correct "We performed quantitative reverse transcription PCR (RT-qPCR)". At best what was performed here was semi-quantitative analysis since relative and not absolute values of the genes was assessed.
3. This statement does not seem to be what the authors intended. It looks like it should read "the molecular dysfunction was corrected/reversed", not restored.

Reviewer #2 (Remarks to the Author):

the authors present an interesting way of providing gene editing and gene replacement. I would encourage them to consider the following points

1. The unmet need of current AAV replacement in humans is unclear as there is no report showing a progressive loss of efficacy after injection of GT in humans. In addition- the amount of SMN protein expressed during life decreases dramatically with age. (see the work of Ramos et al. JCI 2019). In addition- the use of risdiplam or nusinersen could help in patients for which the efficacy of GT would decrease with age.
2. Although it seems to be a difference between the treatment groups- the treatment effect is not more impressive than in the first papers of the Barkat's group or of the Kaspar's group (although this paper has been retracted since then). In this context- I would encourage the authors to benchmark their construct against the established constructs - so that the reader may appreciate what is the actual added value and if the group treated with the gene replacement only benchmarks with existing data.
3. In this context- and given the fact that current treatments are mostly limited by age at

administration- the translation of the findings into a human products sounds really difficult. I would encourage the authors to tempered the statements about potential translation

4. n values should be indicated on the KM curve and the weight curve as I suspect that the end of the curve accounts is supported by a very llimited number of animals.

5. Authors claim that they have no competing interest- but the 2 first authors have an affiliation with the commercial entity. Can they confirm that there is no link between the approaches that is presented here and their company.

Minor

Frequency of SMA estimated by merging all NBS program available is 1:14,848 (rather than approx 1/11000)

Authors claim that AAV is safe- which could be slightly tempered. The exemple of X-MTM is probably really not appropriate to support this statement (4 patients died following GT). Also recent report in the NEJM of a patient who died in DMD + systematic review of safety that reveals several fatal cases.

Please avoid using drug commercial name (zolgensma)

"Recently- Mendell and al. showed..." The paper was published 6 years ago- it is not that recent

Reviewer #3 (Remarks to the Author):

Nature Communications. NCOMMS-23-37948

"Therapeutic strategy for spinal muscular atrophy by combining gene supplementation and genome editing"

These authors present a novel approach to increasing SMN protein expression in a mouse model of spinal muscular atrophy by combining a Smn1 cDNA replacement (using AAV9 with a CMV promoter) and gene editing using a CRISPR/Cas9 construct (using NHEJ). The mice are characterized and compared to WT and untreated animals and comparison is made between the cDNA and dual treated groups. The results were mixed, but in some cases demonstrated greater benefit in the combined treatment group.

Comments:

Major:

1. Abstract and Introduction: Please explain why further improvement in the clinical response to a SMN-targeted drug is “highly challenging”. There may be room for additional improvement by using a new technology, but simply having a gene editing strategy does not equate to filling a large unmet need. This should be addressed more fully. It is not correct to state that AAV genomes within episomes are diluted over time – there is no evidence for this in post-mitotic cells such as motor neurons. The authors fail to discuss if and how the current treatment strategies fall short of optimizing restoration of full-length SMN, either by inefficient transduction and/or incomplete target engagement of motor neurons. In addition, the premises for potential in vivo benefit of gene editing using NHEJ remain speculative.

2. Results: the authors are overly simplistic in stating that AAVs are “one of the safest” means for delivery of therapeutic genes, when giving examples of X-MTM (with 5 deaths in the sentinel trial).

3. Results: no data or statistical analyses are presented in this section and are limited to the legends of the figures. Conclusions are stated without the data to support them. Positive results are emphasized.

Minor:

1. The English grammar and wording could be improved in numerous instances. For example, in the Abstract, “motor nerve cells” would be more precisely stated as “motor neurons” as is used in the Introduction. The summary of how ASOs work is awkward and gives the wrong impression that it restores normal splicing and full-length mRNA in all cases rather than increasing by about 2 to 3-fold. Wrong words are sometimes used.

2. Discussion: Suggest using the generic name for Zolgensma in the manuscript (onasemnogene abeparvovec). Approval by FDA was in 2019, so not so recent. Efficacy has been shown up to 7 years in a small number of children, and over 1000 by my estimate to be 3-Years post treatment without a reported decline in clinical effect – so durability of effect seems quite good.

Reviewer #1

In this study the authors sought to determine if a CRISPR-Cas9 based homology-independent targeted integration strategy that targets the mouse *smn* gene in combination with delivery of human SMN1 would improve the SMA disease phenotype using a severe SMA mouse model. This strategy would be beneficial over the approved gene therapy strategy which suffers from a dilution effect over time in dividing cells. While the SMA gene therapy treatment has dramatically changed the disease landscape, treated patients have shown signs of motor decline within 6 months to a year after treatment, requiring additional treatment with the splicing modifiers. Therefore, alternative strategies to provide more stable induction of SMN is warranted. The studies outlined here show promising results to this end, but there remain areas of concern that should be addressed before this study could be considered suitable for publication.

Major Concerns:

1. The language used throughout the manuscript is very imprecise. The manuscript requires extensive editing, particularly in the results section. Statements like "look very healthy" does not reflect quantitative assessment of improved disease phenotype.

Thank you so much for your constructive feedback. We appreciate your insights into the language precision in our manuscript. We understand the importance of clarity and quantitative representation, and we have made the necessary revisions to address these concerns. Your input is invaluable in refining the manuscript, and we are committed to ensuring a more accurate and precise presentation of our findings.

2. The finding that AAV-PHP.eB was more efficient in the spinal cord and brain than AAV9 is not substantiated by the data resented here since there was only visual qualitative assessment and no quantification of that data.

We appreciate the reviewer for pointing out this issue. We recognize the importance of providing quantitative data to substantiate our findings regarding the relative efficiency of AAV-PHP.eB and AAV9 in the spinal cord and brain. We have now performed a rigorous quantitative assessment using real-time quantitative reverse transcription PCR. The results for spinal cords and livers will be incorporated into the revised manuscript

(Fig. 1c, d). The result for the brain showed more *Gfp* induction by AAV-PHP.eB compared to AAV9 but was not statistically significant (male $p=0.1252$, female $p=0.0571$, as shown in the attached figure below). We appreciate your guidance, and this enhancement will significantly strengthen the scientific validity of our study.

3. Given that that DUET strategy was employed because of the potential lag in inducing SMN levels inherent in the HITI-mediated gene correction strategy, it would be pertinent to compare the difference in SMN protein levels across the tissues examined. Outside of some gene expression data in Supplementary Figure 6 there is nothing in this study that demonstrates persistent improvement in SMN levels, which was the central hypothesis.

Thank you for your insightful comments and suggestions. We appreciate the opportunity to address your concerns and provide additional clarity on the important aspects of our study. We acknowledge the importance of comparing SMN1 protein levels in the brains and spinal cords. In response to your valuable feedback, we have incorporated new Western blot data into the manuscript, allowing for a direct assessment of SMN1 protein levels.

This new data, presented in Fig. 3g, demonstrates distinct bands corresponding to endogenous mouse SMN1 in untreated WT mice, which were not detected in untreated SMA mice. The band of slightly higher molecular weight in SMA mice treated with cDNA and DUET indicates exogenous SMN1 (40 kDa) because we included Myc-tag and FLAG-tag at the N-terminal of exogenous mSmn. Importantly, in SMA mice treated with

DUET, the SMN1 band appeared at a position consistent with endogenous mouse SMN1 (37 kDa), suggesting HITI-mediated SMN1 protein expression. We believe that these additions strengthen the manuscript by providing a more comprehensive analysis of SMN1 protein levels and supporting the central hypothesis of our study.

4. The sex differences observed in the DUET treated animals are potentially interesting but this needs to be reconciled since there are not many sex differences that have been reported in SMA disease presentation or response to treatment. The only other sex difference, this reviewer can remember, is in the milder SMN2b- mouse, not the SMN delta7 mice used in this study.

We appreciate the reviewer's insightful comments regarding the observed sex differences in our study. As correctly noted, sex-related variations in SMA disease presentation or response to treatment have not been extensively reported in the literature, especially in models utilizing the SMN delta7 mice, which we employed in our study. The intriguing outcomes indicating better therapeutic responses in female SMA mice treated with ASO demonstrate better therapeutic outcomes in females within mild SMA mouse models (Howell et al., Mol. Ther., 2017; Deguise et al., EBioMedicine, 2020). While these findings indeed warrant careful consideration, we acknowledge the scarcity of comprehensive investigations into sex-dependent effects on SMA pathogenesis or treatment response. To address the reviewer's valid concern and explore potential explanations, we are carefully considering several factors that may contribute to the observed sex differences. One plausible factor is the size disparity between female and male SMA mice at the time of AAV injection along with developmental differences, leading to variations in AAV density as illustrated in Supplementary Figure 2c and Figure 5b. We hypothesize that the higher AAV density in female SMA mice, despite receiving the same AAV dosage, could influence treatment efficiency. We have incorporated a detailed discussion of this hypothesis in the revised discussion section.

We believe that a more nuanced exploration of these sex-related nuances will not only contribute to our understanding of SMA but also shed light on potential factors influencing treatment responses. We will ensure to thoroughly address and reconcile these observations in our revised manuscript, providing the necessary context and

acknowledging the complexity of sex-related variables in SMA research. We hope this clarification adequately addresses the reviewer's concerns.

Your guidance is greatly appreciated, and we are committed to enhancing the clarity and depth of our analysis.

Minor Concerns:

1. Rats(rodents) have a single copy of the SMN gene. It's therefore unclear what the authors mean by "we removed the homologous sequence on the donor by incorporating a portion of rat intron 1 of Snn1 including the splicing acceptor, codon-optimized mouse Snn1 cDNA"

Thank you for your careful review and thoughtful comments. We appreciate your attention to detail. Concerning your inquiry about the statement "we removed the homologous sequence on the donor by incorporating a portion of rat intron 1 of Snn1 including the splicing acceptor, codon-optimized mouse Snn1 cDNA," we would like to clarify that the term "homologous sequence" refers to identical sequences between the donor sequence and the intron 1 of Snn1 in the mouse genome. These similarities were observed to potentially lead to unexpected recombination events, as previously documented (Suzuki et al, Cell Res, 2019). To ensure proper splicing following the knock-in, it was crucial to include a functional splicing acceptor sequence ahead of the inserted exon (i.e., codon-optimized mouse Snn1 cDNA) within the donor. Thus, to eliminate the homologous sequence and introduce the splicing acceptor site, we engineered a portion of rat intron 1 of Snn1 (inclusive of the splicing acceptor) preceding the codon-optimized mouse Snn1 cDNA on the donor construct. We apologize if this was not explicitly clear in the manuscript, and we have made the necessary revisions to provide a more detailed explanation of this process, as reflected in the main text on page 4.

2. This statement is not factually correct "We performed quantitative reverse transcription PCR (RT-qPCR)". At best what was performed here was semi-quantitative analysis since relative and not absolute values of the genes was assessed.

We appreciate the careful consideration of our methods. We would like to clarify that our study employed real-time quantitative reverse transcription PCR (qRT-PCR) techniques

for the analysis of RNA expression levels. In our methodology, RNA was converted to cDNA, and subsequent real-time PCR analyses were conducted using the CFX384 Touch Real-Time PCR Detection System. This approach allowed us to precisely quantify the expression levels of our genes of interest. We have incorporated “Real-time quantitative reverse transcription PCR (qRT-PCR)” in the main text. We hope this clarification addresses the concern regarding the quantitative nature of our analyses.

3. This statement does not seem to be what the authors intended. It looks like it should read “the molecular dysfunction was corrected/reversed“, not restored.

Thank you for your careful examination of our manuscript, and we appreciate your insightful comments. We acknowledge your suggestion regarding the terminology used to describe the impact of our treatments on molecular dysfunction. You are correct; a more accurate representation of our findings would be to state that "the molecular dysfunction was corrected/reversed" rather than "restored." We have made the necessary revisions to offer a more accurate portrayal, as reflected in the main text on page 5.

Reviewer #2

the authors present an interesting way of providing gene editing and gene replacement. I would encourage them to consider the following points

1. The unmet need of current AAV replacement in humans is unclear as there is no report showing a progressive loss of efficacy after injection of GT in humans. In addition- the amount of SMN protein expressed during life decreases dramatically with age. (see the work of Ramos et al. JCI 2019). In addition- the use of risdiplam or nusinersen could help in patients for which the efficacy of GT would decrease with age.

While we acknowledge the efficacy of current SMN1 supplementation therapies, including onasemnogene abeparvovec-xioi (Zolgensma), it's essential to note the existing limitations observed in both our mouse experiments and ongoing evaluations in human SMA patients. Recent reports have indicated that onasemnogene abeparvovec-xioi upregulates SMN transcript levels by approximately 25% in the spinal cord (Nat. Med. 2021 27, 1701–1711), which falls short of achieving complete rescue. And in human SMA patients, the long-term effects are still under evaluation.

Your mention of the work by Ramos et al. (JCI 2019) is duly noted. We acknowledge that the amount of SMN protein expressed during the lifespan decreases, and this could have implications for the long-term efficacy of gene therapy. However, it's crucial to emphasize that the initial SMN expression is vital for SMA therapy. In this revision, we have included Western blot analysis data (please see the answer to the 2nd question). As reported in Nat. Neurosci, 930-934, 2021, AAV is more stable in neurons than in proliferative cells. However, our data, as presented in Fig. 5e, indicates stable genome-editing rates in the spinal cord and an increase in the liver at P40 weeks compared to P20 weeks. The increase in the liver may be attributed to the proliferation of genome-edited cells. These findings align with the concept that stable SMN1 expression, sustained by HITI-mediated SMN1, is crucial for both motor neurons and peripheral organs, providing a foundation for the observed additional therapeutic benefits with the Gene-DUET strategy. We hope these clarifications address your concerns, and we have incorporated this explanation in the revised discussion section.

2. Although it seems to be a difference between the treatment groups- the treatment effect is not more impressive than in the first papers of the Barkat's group or of the Kaspar's group (although this paper has been retracted since then). In this context- I would encourage the authors to benchmark their construct against the established constructs - so that the reader may appreciate what is the actual added value and if the group treated with the gene replacement only benchmarks with existing data.

We appreciate the reviewer's suggestion to benchmark our construct against established ones in the field. While direct comparisons between different studies are challenging due to variations in experimental conditions, we would like to present a comprehensive overview of median survival data for various therapeutic strategies in SMA mice, as outlined in the supplementary Data 7. The data reveal that single-strand AAV is 40% less efficient than self-complementary AAV, and intracerebroventricular (ICV) injection of AAV is more effective than systemic injection. Notably, the median survival reported for self-complementary AAV9-SMN by Dominguez et al. in 2011 and by Besse et al. in 2020 is 199 days and 142 days, respectively. Additionally, the current strategy involving a combination of base-editing of SMN2 and nusinersen shows a median survival of 77 days. Our data demonstrate that, even with the acknowledged lower efficiency of single-strand AAV, the Gene-DUET achieved a remarkable median survival of 201 days, surpassing the 124 days observed with only SMN cDNA in mixed-sex results. This outcome strongly supports the conclusion that our strategy is more efficient for survival in SMA mice.

Furthermore, the new Western blot analysis presented in Fig. 3g highlights the increase in SMN1 protein levels resulting from our strategy, with contributions from both cDNA supplementation and HITI-mediated genome editing. This effect approaches endogenous SMN1 levels in the spinal cords, suggesting potential additional therapeutic benefits.

We have made sure to emphasize these comparative outcomes more explicitly in the manuscript, providing a clear perspective on the advancements and superior performance of the Gene-DUET strategy over existing approaches.

3. In this context- and given the fact that current treatments are mostly limited by age at administration- the translation of the findings into a human products sounds really difficult. I would encourage the authors to tempered the statements about potential translation

We appreciate the reviewer's insight into the challenges associated with translating our findings into human applications. It's essential to acknowledge the current limitations of AAV, including size capacity and safety concerns, which we recognize as areas needing further development. We concur that early intervention in SMA is crucial for treatment success, and we recognize the complexities of translating our Gene-DUET strategy to human applications. While we are encouraged by the beneficial effects observed in our SMA mouse model, we understand the importance of tempering statements about the potential translation of our strategy. We agree that the current treatments are often limited by the age at administration, and the translation of our findings into a clinical setting requires careful consideration of various factors, including safety, efficacy, and practicality. We acknowledge that further preclinical studies in other animal models, including non-human primates, are essential prerequisites for advancing toward clinical applications. We value the reviewer's perspective, and in the revised manuscript, we have refined our statements to appropriately reflect the current stage of our research and the considerations involved in potential clinical applications.

4. n values should be indicated on the KM curve and the weight curve as I suspect that the end of the curve accounts is supported by a very limited number of animals.

We appreciate the reviewer's attention to the representation of experimental numbers in our study. The mice numbers are indeed included in Figure 5g for Kaplan-Meier survival curves and weight curves used in our experiments. Additionally, we have incorporated a new supplementary Data that includes our raw data related to figures. This addition aims to enhance the accessibility of our data, allowing for a more in-depth examination and scrutiny of our findings. We believe that this comprehensive presentation ensures the clarity and validity of our results.

5. Authors claim that they have no competing interest- but the 2 first authors have an affiliation with the commercial entity. Can they confirm that there is no link between the approaches that is presented here and their company.

We appreciate the reviewer's diligence in examining potential conflicts of interest. The authors and corresponding author are currently employed by, and are shareholders in, Altos Labs, Inc. We have confirmed unequivocally that there is no conflict of interest that could bias the outcomes or interpretation of the results presented in this study. The research was primarily conducted in Salk Institute, and the work in the commercial entity was limited to aspects unrelated to the data and findings reported here. Our commitment to scientific integrity and transparency remains paramount.

Minor

Frequency of SMA estimated by merging all NBS program available is 1:14,848 (rather than approx 1/11000)

We appreciate the reviewer's attention to the frequency estimation of SMA. We've updated the frequency of SMA and included the relevant reference in the main text on page 2.

Authors claim that AAV is safe- which could be slightly tempered. The exemple of X-MTM is probably really not appropriate to support this statement (4 patients died following GT). Also recent report in the NEJM of a patient who died in DMD + systematic review of safety that reveals several fatal cases.

We appreciate the reviewer's attention to the safety claims made in the manuscript. We acknowledge that the field of gene therapy has witnessed specific instances that underscore the importance of nuanced language regarding safety. In light of this, we have revised the manuscript to remove the term "safe" in association with AAV, opting for more cautious language that reflects the evolving understanding of AAV-based therapies and the challenges posed by individual cases such as those mentioned, notably in X-MTM and DMD.

Please avoid using drug commercial name (zolgensma)

We appreciate the reviewer's suggestion to avoid the use of commercial drug names, and we have carefully addressed this concern in the revised version of the manuscript. In place of the commercial name "Zolgensma," we now use the generic name "onasemnogene abeparvovec-xioi" to enhance clarity and precision. We believe this modification aligns with the reviewer's recommendation, and we thank them for their valuable input.

"Recently- Mendell and al. showed..." The paper was published 6 years ago- it is not that recent

We appreciate the reviewer's attention to detail and acknowledge that the paper by Mendell et al. has been available for a significant duration. We have now revised the manuscript to accurately reflect the publication date and avoid any potential misrepresentation.

Reviewer #3 (Remarks to the Author):

Nature Communications. NCOMMS-23-37948

“Therapeutic strategy for spinal muscular atrophy by combining gene supplementation and genome editing”

These authors present a novel approach to increasing SMN protein expression in a mouse model of spinal muscular atrophy by combining a *Smn1* cDNA replacement (using AAV9 with a CMV promoter) and gene editing using a CRISPR/Cas9 construct (using NHEJ). The mice are characterized and compared to WT and untreated animals and comparison is made between the cDNA and dual treated groups. The results were mixed, but in some cases demonstrated greater benefit in the combined treatment group.

Comments:

Major:

1. Abstract and Introduction: Please explain why further improvement in the clinical response to a SMN-targeted drug is “highly challenging”. There may be room for additional improvement by using a new technology, but simply having a gene editing strategy does not equate to filling a large unmet need. This should be addressed more fully. It is not correct to state that AAV genomes within episomes are diluted over time – there is no evidence for this in post-mitotic cells such as motor neurons. The authors fail to discuss if and how the current treatment strategies fall short of optimizing restoration of full-length SMN, either by inefficient transduction and/or incomplete target engagement of motor neurons. In addition, the premises for potential in vivo benefit of gene editing using NHEJ remain speculative.

We appreciate the reviewer's insightful comments and the opportunity to address these crucial points. While we acknowledge the efficacy of current SMN1 supplementation therapies, including onasemnogene abeparvovec-xioi (Zolgensma), it's essential to note the existing limitations observed in both our mouse experiments and ongoing evaluations in human SMA patients. Recent reports have indicated that onasemnogene abeparvovec-xioi upregulates SMN transcript levels by approximately 25% in the spinal cord (Nat. Med. 2021 27:10 27, 1701–1711, 2021), which falls short of achieving complete rescue.

And in human SMA patients, the long-term effects are still under evaluation. Direct comparisons across studies can be challenging due to variations in experimental conditions. Still, we'd like to provide summary of median survival data by therapeutic strategy in SMA mice, as outlined in the supplementary Data 7. Our data demonstrate that even with the acknowledged lower efficiency of single-strand AAV, Gene-DUET achieved a median survival of 201 days, surpassing the 124 days observed with only SMN cDNA in mixed-sex results. This outcome strongly supports the conclusion that our strategy is more efficient than conventional strategy for survival in SMA mice.

The new Western blot analysis in Fig. 3g highlights the increase in SMN1 protein levels resulting from our strategy, with contributions from both cDNA supplementation and HITI-mediated genome editing. This effect approaches endogenous SMN levels in the spinal cords, suggesting potential additional therapeutic benefits.

*Regarding the question about the dilution of AAV in post-mitotic cells, we appreciate the opportunity to clarify these points in the revised manuscript. As the reviewer correctly pointed out, AAV stability in neurons, as reported in *Nat. Neurosci*, 930-934, 2021, is more pronounced than in proliferative cells. However, our data, as presented in Fig. 5e, indicates stable genome-editing rates in the spinal cord and an increase in the liver at P40 weeks compared to P20 weeks. This may be attributed to the proliferation of genome-edited cells. These findings align with the concept that stable SMN expression, sustained by HITI-mediated SMN, is crucial for both motor neurons and peripheral organs, providing a foundation for the observed additional therapeutic benefits with the Gene-DUET strategy.*

2. Results: the authors are overly simplistic in stating that AAVs are “one of the safest” means for delivery of therapeutic genes, when giving examples of X-MTM (with 5 deaths in the sentinel trial).

We appreciate the reviewer's attention to the safety claims made in the manuscript. We acknowledge that the field of gene therapy has witnessed specific instances that underscore the importance of nuanced language regarding safety. In light of this, we have revised the manuscript to remove the term "safe" in association with AAV, opting for

more cautious language that reflects the evolving understanding of AAV-based therapies and the challenges posed by individual cases such as those mentioned, notably in X-MTM.

3. Results: no data or statistical analyses are presented in this section and are limited to the legends of the figures. Conclusions are stated without the data to support them. Positive results are emphasized.

We appreciate the reviewer's insightful comments and acknowledge the importance of providing robust statistical analyses to support our conclusions. In response to this concern, we have taken diligent steps to include statistical analyses in the main text and the legends, providing a more transparent and evidence-backed presentation of our findings. We have also incorporated a new supplementary Data that includes our raw data related to each figure. This addition aims to enhance the accessibility of our data, allowing for a more in-depth examination and scrutiny of our findings. Your guidance is greatly appreciated, and we are committed to enhancing the clarity and depth of our analysis.

Minor:

1. The English grammar and wording could be improved in numerous instances. For example, in the Abstract, “motor nerve cells” would be more precisely stated as “motor neurons” as is used in the Introduction. The summary of how ASOs work is awkward and gives the wrong impression that it restores normal splicing and full-length mRNA in all cases rather than increasing by about 2 to 3-fold. Wrong words are sometimes used.

Thank you so much for your constructive feedback. We appreciate your insights into the language precision in our manuscript. We understand the importance of clarity and quantitative representation, and we have made the necessary revisions to address these concerns. Additionally, we acknowledge the need for precision in describing the mechanism of action of ASOs and will revise the summary accordingly to accurately reflect their impact on splicing and mRNA levels.

2. Discussion: Suggest using the generic name for Zolgensma in the manuscript (onasemnogene abeparvovec). Approval by FDA was in 2019, so not so recent. Efficacy

has been shown up to 7 years in a small number of children, and over 1000 by my estimate to be 3-Years post treatment without a reported decline in clinical effect – so durability of effect seems quite good.

We appreciate the suggestion to use the generic name, onasemnogene abeparvovec-xioi, in the manuscript, and we have made the necessary changes accordingly. Regarding the timing of FDA approval, we acknowledge that it occurred in 2019. We've updated the text to reflect this accurately. Additionally, we have included recent data on the efficacy of onasemnogene abeparvovec-xioi, noting that its effectiveness has been demonstrated for up to 7 years in a small cohort of children and for over 3 years in a larger group without a reported decline in clinical effect.

REVIEWER COMMENTS

Reviewer #1 (Remarks to the Author):

The concerns raised by this reviewer have been addressed and resolved in the revised manuscript submission, improving its completeness and quality.

Minor point: The authors should correct terms like "SMN1 protein" in text and figures. *SMN1* is the gene. SMN is the protein. Also mice only have one SMN gene so any indication of mouse *smn1* (*mSmn1*) is incorrect.

Reviewer #2 (Remarks to the Author):

The authors have addressed my comments. Congratulation for this nice paper

Reviewer #3 (Remarks to the Author):

Nature Communications manuscript NCOMMS-23-37948A

"Therapeutic strategy for spinal muscular atrophy by combining gene supplementation and genome editing"

These investigators have responded to the reviewers' critique of their initial manuscript with a significant revision. They describe with greater clarity their hypothesis and experimental data using a dual CRISPR-Cas9 (using a NHEJ strategy) and *sMN1* cDNA gene transfer strategy to upregulate SMN expression in a SMA mouse model. The construct they developed included a melding of rat intron 1 and 3'UTR and mouse exons 2-8. SMN cDNA knock-in was successfully achieved following systemic delivery of AAV-PHP.eB-SMN1-HITI and AAV-PHP.eB-Cas9, administered via facial vein injectin at P0.5. This combined strategy was demonstrated to increase SMN expression in the d7 mouse spinal cord and partial rescue with prolonged survival and improved motor function. Females did better than males, attributed to having received a higher dose/g body weight.

Comments:

1. Introduction:

a. The comment that “gene supplemental therapy cannot permanently restore endogenous SMN1 expression...” is not entirely correct. Transduction of SMN1 cDNA into a nuclear episome within motor neurons has been demonstrated to have substantial durability to date, of at least 8 years in humans and longer in animals. There are no data to indicate a deterioration over time. The authors acknowledge this in the Discussion, so there is a lack of alignment on this topic.

b. The safety profile is also more favorable with an episome than with an integrating vector. The risk of integration into the germline was not addressed effectively by the authors. Only the potential benefits were emphasized.

c. As ASO therapy for SMA with nusinersen is delivered intrathecally penetration of the blood-brain barrier is not an issue.

d. It is presumptuous to state: “in situ gene correction of the SMN1 mutation becomes imperative for the permanent treatment of all SMA patients.” The extent of motor neuron transduction is important. Type 1 patients have different needs for SMN restoration from types 2 and 3, as discussed by the authors. Rescue of SMA with upregulation of SMN is not a one-size-fits-all situation.

2. Wild-type cDNA supplementation was achieved by overexpressing mouse *Smn1* cDNA.

While survival and motor function was improved at 20 and 40 weeks observation, the authors fail to acknowledge the active discussion within the SMA gene therapy community whether there is a risk of toxic over-expression of SMN (see Van Alstyne et al, ref 30). This is a critical topic when a strategy to increase SMN expression is being addressed, as with the central hypothesis here. There is also an increased risk of liver toxicity. Cellular motor neuron and DRG pathology and immunohistochemistry studies were not performed to address this topic.

3. These investigators demonstrated that they could successfully utilize HITI-mediated genome editing to partially rescue the phenotype in the severe d7 mouse model of SMA, but with survival increased only by 2 to 3.5 days (~20%) as compared to the three drugs approved by regulatory authorities with much more robust responses of 10+ fold increase in survival. This was attributed to late timing for administration rather than poor efficiency. The latter needs to be considered and addressed.

4. DUET-treated d7 mice had improved survival and motor function at 2 weeks but exhibited

poor systemic protection as ear and digit necrosis and shorter tails were identified at 5 weeks. These short-term observations and significant limitations are hardly dramatic improvements, as claimed by the authors.

5. Some molecular correction was demonstrated using RNAseq of spinal cord tissue from cDNA and DUET treated mice, but not with HITI alone. Partial correction is not reversal, as claimed by the authors.

6. Weight gain is no different in the cDNA and DUET treated SMA mice and plateaus early, diverging from hets (Fig 5). This is not persuasive of a treatment effect despite survival to >300d in the females.

7. Nor did there appear to be a benefit in motor activity (fig S5) with the DUET strategy over the cDNA.

Reviewer #1:

The concerns raised by this reviewer have been addressed and resolved in the revised manuscript submission, improving its completeness and quality.

We are glad to hear that this reviewer found the revisions satisfactory and that they have contributed to enhancing the completeness and quality of the manuscript.

Minor point: The authors should correct terms like "SMN1 protein" in text and figures. *SMN1* is the gene. SMN is the protein. Also mice only have one SMN gene so any indication of mouse *smn1* (*mSmn1*) is incorrect.

*We appreciate the reviewer's attention to detail and additional request for clarification regarding the terminology used. We have corrected instances of "SMN1 protein" to "SMN1" in both the text and figures, per the reviewer's suggestion. Thank you for bringing this to our attention. Regarding the indication of "mouse smn1," while it's true that mice only have one SMN gene, the official symbol is *Smn1*. We believe that indicating "mouse *Smn1*" instead of *mSmn1* provides clarity for readers and this correction has been made in the text.*

Reviewer #2:

The authors have addressed my comments. Congratulations for this nice paper

Thank you very much for this reviewer's positive feedback and acknowledgment of our efforts in addressing their comments. We greatly appreciate their support and encouragement.

Reviewer #3 (Remarks to the Author):

Nature Communications manuscript NCOMMS-23-37948A "Therapeutic strategy for spinal muscular atrophy by combining gene supplementation and genome editing"

These investigators have responded to the reviewers' critique of their initial manuscript with a significant revision. They describe with greater clarity their hypothesis and experimental data using a dual CRISPR-Cas9 (using a NHEJ strategy) and SMN1 cDNA gene transfer strategy to upregulate SMN expression in a SMA mouse model. The construct they developed included a melding of rat intron 1 and 3'UTR and mouse exons 2-8. SMN cDNA knock-in was successfully achieved following systemic delivery of AAV-PHP.eB-SMN1-HITI and AAV-PHP.eB-Cas9, administered via facial vein injection at P0.5. This combined strategy was demonstrated to increase SMN expression in the d7 mouse spinal cord and partial rescue with prolonged survival and improved motor function. Females did better than males, attributed to having received a higher dose/g body weight.

Comments:

1.Introduction:

a. The comment that “gene supplemental therapy cannot permanently restore endogenous SMN1 expression...” is not entirely correct. Transduction of SMN1 cDNA into a nuclear episome within motor neurons has been demonstrated to have substantial durability to date, of at least 8 years in humans and longer in animals. There are no data to indicate a deterioration over time. The authors acknowledge this in the Discussion, so there is a lack of alignment on this topic.

We appreciate the reviewer's insights and the opportunity to further clarify our statements. Indeed, exogenous SMN delivered by AAV remains stable in motor neurons but is subject to dilution in the liver. The presence of SMN in the liver underscores its importance in peripheral tissues. Our deep sequence data indicated stable genome-editing rates in the spinal cord and an increase in the liver, which can retain the endogenous SMN expression. Additionally, our emphasis was on highlighting the potential benefits of genome editing to restore endogenous SMN expression, which contrasts with exogenous gene supplementation. The distinction lies in the ability of genome editing to restore SMN expression under endogenous control mechanisms, potentially offering a safer alternative to exogenous supplementation. To provide clarity for readers, we have revised the statement to read: 'this gene supplementation therapy cannot assure permanent restoration of endogenous SMN1 expression'.

b. The safety profile is also more favorable with an episome than with an integrating vector. The risk of integration into the germline was not addressed effectively by the authors. Only the potential benefits were emphasized.

We appreciate the reviewer's thoughtful consideration of the safety aspects related to gene therapy vectors. Our focus was primarily on highlighting the potential benefits of utilizing episomal vectors, such as AAV, to restore gene expression in a transient and non-integrating manner. It's crucial to acknowledge that, in general, AAV vectors have a favorable safety profile due to their episomal nature, which minimizes the risk of genomic integration and subsequent germline transmission. However, we recognize the importance of addressing the potential risks associated with vector integration, which can alter the genotoxicity profile. While the studies we referenced indicated that germline transmission of certain AAV vectors is unlikely (Gene Therapy, volume 30, 581-586, 2023), we acknowledge that further research is necessary to comprehensively assess the safety of AAV-based gene therapies, especially concerning long-term effects and potential integration events. We have incorporated these considerations into the revised Discussion section to provide a balanced perspective on the safety implications of gene therapy using AAV vectors.

c. As ASO therapy for SMA with nusinersen is delivered intrathecally penetration of the blood-brain barrier is not an issue.

We appreciate the reviewer's comment regarding the delivery of ASO therapy, particularly nusinersen, which is administered intrathecally. Indeed, intrathecal administration bypasses the blood-brain barrier, allowing direct access to the central nervous system. We have incorporated the consideration that ASOs face challenges such as the requirement for repetitive administrations into the revised Introduction section.

d. It is presumptuous to state: “in situ gene correction of the SMN1 mutation becomes imperative for the permanent treatment of all SMA patients.” The extent of motor neuron transduction is important. Type 1 patients have different needs for SMN restoration from types 2 and 3, as discussed by the authors. Rescue of SMA with upregulation of SMN is not a one-size-fits-all situation.

We appreciate the reviewer's insightful comments and acknowledge the complexity surrounding the treatment of SMA. Our statement regarding in situ gene correction of the SMN1 mutation was intended to highlight a potential future direction for SMA therapy. We recognize that SMA is primarily caused by SMN1 gene deficiency; if the SMN1 gene is normal and functioning properly, SMA would not typically occur. Therefore, we believe that the success of genome correction for the SMN1 mutation can provide permanent treatment for all SMA patients. However, we agree it was presumptuous to state this definitively. We have revised the statement to emphasize that 'the success of in situ gene correction of the SMN1 mutation holds the potential to provide a permanent cure for all SMA patients' in the revised manuscript.

2. Wild-type cDNA supplementation was achieved by overexpressing mouse *Smn1* cDNA. While survival and motor function was improved at 20 and 40 weeks observation, the authors fail to acknowledge the active discussion within the SMA gene therapy community whether there is a risk of toxic over-expression of SMN (see Van Alstyne et al, ref 30). This is a critical topic when a strategy to increase SMN expression is being addressed, as with the central hypothesis here. There is also an increased risk of liver toxicity. Cellular motor neuron and DRG pathology and immunohistochemistry studies were not performed to address this topic.

We appreciate the reviewer's valuable comments and the opportunity to discuss the potential risks associated with SMN overexpression, as highlighted by Van Alstyne et al. (ref 30). Our study utilized SMA mice and focused primarily on assessing survival outcomes following cDNA supplementation or utilization of the Gene-DUET strategy. While we acknowledge the concerns raised regarding the potential for toxic overexpression of SMN and liver toxicity, we believe that the benefits observed in our study outweigh the potential risks. This underscores our belief in the potential of genome editing to provide endogenous levels of SMN expression. In our study, we prioritized survival outcomes to evaluate the therapeutic impact of the Gene-DUET strategy. We also demonstrated stable gene correction efficiency between 20 weeks old and 40 weeks old, suggesting that the AAV-infected cells are still viable. We recognize the importance of conducting comprehensive toxicity studies, including cellular motor neuron and DRG pathology, and immunohistochemistry analyses to address these concerns. However, due to the limitations of available samples, we were unable to perform these analyses in the current study. Future investigations will incorporate these analyses to provide a more comprehensive assessment of the safety profile of our treatment approach. We have incorporated these considerations into the revised Discussion section to provide a balanced interpretation of our findings and to acknowledge the areas requiring further

investigation, particularly regarding the potential risks associated with SMN overexpression and liver toxicity. Thank you for highlighting these critical aspects.

3. These investigators demonstrated that they could successfully utilize HITI-mediated genome editing to partially rescue the phenotype in the severe d7 mouse model of SMA, but with survival increased only by 2 to 3.5 days (~20%) as compared to the three drugs approved by regulatory authorities with much more robust responses of 10+ fold increase in survival. This was attributed to late timing for administration rather than poor efficiency. The latter needs to be considered and addressed.

We appreciate the reviewer's insightful comments and the opportunity to discuss our Gene-DUET strategy. Our study indeed demonstrated a modest increase in survival in SMA mice following one-time AAV-SMN1-HITI administration at postnatal 0.5 day, which underscores the rationale behind developing the Gene-DUET strategy. While it is intriguing to consider administering the AAVs in utero, it was not feasible to address this aspect in our study due to constraints in technical proficiency. It's important to note that our study focused on assessing HITI-mediated genome editing as part of a combinatorial treatment strategy (Gene-DUET), which includes cDNA supplementation. This approach was designed to harness the benefits of both genome editing and gene supplementation to address the multifaceted challenges associated with SMA. Gene-DUET achieved longer survival compared to SMA mice treated with HITI alone or cDNA alone, highlighting its potential efficacy.

4. DUET-treated d7 mice had improved survival and motor function at 2 weeks but exhibited poor systemic protection as ear and digit necrosis and shorter tails were identified at 5 weeks. These short-term observations and significant limitations are hardly dramatic improvements, as claimed by the authors.

We appreciate the reviewer's insights and the opportunity to address the observations related to tail necrosis in our study. The frequency of tail necrosis observed in both cDNA-treated and DUET-treated SMA mice was not different. Regarding the observations of ear and digit necrosis and shorter tails in treated SMA mice, similar findings were also observed after treatment with ASO and AAV-SMN in previous studies. These observations highlight the need for continued refinement and optimization to address systemic SMN expression and its impact on peripheral tissues. It's essential to emphasize that our study provides data demonstrating improved survival following DUET treatment. However, we acknowledge the significant limitations observed in systemic protection and peripheral tissue outcomes. These findings underscore the ongoing challenges in developing effective gene therapy strategies for SMA and the necessity for further research to optimize treatment outcomes and mitigate potential adverse effects. We have incorporated these considerations into the revised Discussion section to provide a balanced interpretation of our findings and to acknowledge the areas requiring improvement and future research efforts.

5. Some molecular correction was demonstrated using RNAseq of spinal cord tissue from cDNA and DUET treated mice, but not with HITI alone. Partial correction is not reversal, as claimed by the authors.

We appreciate the reviewer's attention to the molecular corrections observed in our study. It is correct that HITI alone did not achieve molecular correction to the same extent as cDNA and DUET treatments, although HITI-treated SMA mice did show better righting ability than untreated SMA mice. This underscores the rationale behind developing the Gene-DUET strategy. Indeed, the RNAseq data from spinal cord tissues demonstrated significant improvements in molecular dysfunction following cDNA and DUET treatments. While it is true that partial correction does not equate to complete reversal, our study focused on the notable improvements in molecular, survival, and behavioral phenotypes observed in SMA mice following DUET treatment. The Gene-DUET approach, by combining cDNA supplementation and HITI-mediated genome editing, was designed to enhance therapeutic outcomes beyond what either approach alone could achieve.

6. Weight gain is no different in the cDNA and DUET treated SMA mice and plateaus early, diverging from hets (Fig 5). This is not persuasive of a treatment effect despite survival to >300d in the females.

We appreciate the reviewer's observation regarding weight gain in cDNA- and DUET-treated SMA mice compared to heterozygous mice. While weight gain did not show a significant difference between treatment groups, it's important to note that our study primarily focused on survival. We hypothesize that the benefits of genome editing by Gene-DUET may become more apparent in later stages of life. We observed that the decline in body weight before mouse death is significantly less in DUET-treated SMA female mice compared to cDNA-treated SMA female mice (see attached Figure a). We believe this difference and survival extension underscores the potential treatment effect of the Gene-DUET strategy.

7. Nor did there appear to be a benefit in motor activity (fig S5) with the DUET strategy over the cDNA.

We appreciate the reviewer's careful examination of our data on motor function, particularly the open field test. While we acknowledge that statistical significance was not achieved in these specific tests, we observed trends suggesting improved performance in certain parameters of the open field test in DUET-treated SMA mice compared to those treated with cDNA alone (see attached Figures b-d). Additionally, we observed better performance in the righting reflex test in DUET-treated SMA mice compared to those treated with cDNA alone (see attached Figure e, as main Figure 3h). We believe that these trends, along with the difference in body weight decline before death as shown in attached Figure a, combined with the notable survival benefits observed with the DUET strategy, support further investigation into the potential improvements associated with our treatment approach.

Attached Figure

a. Body weight changes of cDNA- or DUET-treated SMA mice in the 6 weeks prior to death. In female SMA mice, the body weight decreased 1 to 2 weeks prior to death. cDNA-treated SMA mice lost 10% of their body weight per week, while DUET-treated SMA mice lost less than 5% of their body weight per week. Data were analyzed by two-way repeated-measures ANOVA. Values in graphs are expressed as mean \pm SEM. ****** $p < 0.005$. **b-d.** Total center zone time (b), total jump counts (c), and average velocity (d) in the open field test. Each data point represents the total value over 30 minutes. $n = 4$ for all mice. Data are represented as mean \pm S.D. **e,** Enlarged figure of the righting reflex test in male mice.

REVIEWERS' COMMENTS

Reviewer #3 (Remarks to the Author):

NCOMMS-23-37948B: Therapeutic strategy for spinal muscular atrophy by combining gene supplementation and genome editing

The authors have satisfactorily addressed the reviewers' remains comments. This reviewer has no additional concerns.